# Structural mechanism for inhibition of PP2A-B56α and oncogenicity by CIP2A

Karolina Pavic[1,2], Nikhil Gupta[1,13], Judit Domènech Omella[3,13], Rita Derua[3,4], Anna Aakula[1], Riikka Huhtaniemi[1], Juha A. Määttä[5], Nico Höfflin[6], Juha Okkeri[1], Zhizhi Wang[7], Otto Kauko[1], Roosa Varjus[1], Henrik Honkanen[1], Daniel Abankwa[2], Maja Köhn[6,8], Vesa P. Hytönen[5], Wenqing Xu[7], Jakob Nilsson[9], Rebecca Page[10], Veerle Janssens[3], Alexander Leitner[11] & Jukka Westermarck[1,12]✉

The protein phosphatase 2A (PP2A) heterotrimer PP2A-B56α is a human tumour suppressor. However, the molecular mechanisms inhibiting PP2A-B56α in cancer are poorly understood. Here, we report molecular level details and structural mechanisms of PP2A-B56α inhibition by an oncoprotein CIP2A. Upon direct binding to PP2A-B56α trimer, CIP2A displaces the PP2A-A subunit and thereby hijacks both the B56α, and the catalytic PP2Ac subunit to form a CIP2A-B56α-PP2Ac pseudotrimer. Further, CIP2A competes with B56α substrate binding by blocking the LxxIxE-motif substrate binding pocket on B56α. Relevant to oncogenic activity of CIP2A across human cancers, the N-terminal head domain-mediated interaction with B56α stabilizes CIP2A protein. Functionally, CRISPR/Cas9-mediated single amino acid mutagenesis of the head domain blunted MYC expression and MEK phosphorylation, and abrogated triple-negative breast cancer in vivo tumour growth. Collectively, we discover a unique multi-step hijack and mute protein complex regulation mechanism resulting in tumour suppressor PP2A-B56α inhibition. Further, the results unfold a structural determinant for the oncogenic activity of CIP2A, potentially facilitating therapeutic modulation of CIP2A in cancer and other diseases.

Protein phosphatase 2A (PP2A) is one of the major cellular serine/threonine phosphatases estimated to control up to 60 per cent of all serine/threonine phosphorylation. It is not only a tumour suppressor, based on its negative regulation of pro-survival and proliferation factors[1], but also implicated in a variety of other diseases as well as in cellular physiology[2]. Intricate functionality of PP2A is achieved through its complex structural organization in which a scaffolding (A subunit; PP2A-A or PR65) and catalytic C-subunit (PP2Ac), together forming the

[1]Turku Bioscience Centre, University of Turku and Åbo Akademi University, 20520 Turku, Finland. [2]Cancer Cell Biology and Drug Discovery Group, Department of Life Sciences and Medicine, University of Luxembourg, Luxembourg, Esch-sur-Alzette, Luxembourg. [3]Laboratory of Protein Phosphorylation & Proteomics, Department of Cellular & Molecular Medicine, University of Leuven (KU Leuven), B-3000 Leuven, Belgium. [4]SyBioMa, University of Leuven (KU Leuven), B-3000 Leuven, Belgium. [5]Faculty of Medicine and Health Technology, Tampere University, 33520 Tampere, Finland and Fimlab Laboratories, 33520 Tampere, Finland. [6]Faculty of Biology, Institute of Biology III, University of Freiburg, 79104 Freiburg, Germany. [7]School of Life Science and Technology, ShanghaiTech University, Shanghai, China. [8]Signalling Research Centres BIOSS and CIBSS, University of Freiburg, Freiburg, Germany. [9]The Novo Nordisk Foundation Center for Protein Research, University of Copenhagen, Blegdamsvej 3B, 2200 Copenhagen, Denmark. [10]Department of Chemistry and Biochemistry University of Arizona, Tucson, AZ, USA. [11]Department of Biology, Institute of Molecular Systems Biology, ETH Zurich, 8093 Zurich, Switzerland. [12]Institute of Biomedicine, University of Turku, 20520 Turku, Finland. [13]These authors contributed equally: Nikhil Gupta, Judit Domènech Omella. ✉e-mail: jukwes@utu.fi

core enzyme, organize into heterotrimeric holoenzymes with only one member of the regulatory B-subunit. The B-subunits are split into four structurally unrelated families giving rise to over 60 potential PP2A holoenzymes that each may have different substrate selectivity and cellular function[1]. Thereby, characterization of the mechanisms that regulate PP2A protein complex assemblies are central for comprehensive understanding of PP2A biology, and thereby for understanding of general phosphoproteome regulation[1,3].

Among the PP2A complexes, PP2A trimers with B56 subunits are predominantly human tumour suppressors[4–7]. Through loss-of-function studies it has been shown that PP2A-B56α inhibition drives malignant transformation of human cells, as well promotes malignant growth[6,8]. Recent studies have also linked PP2A-B56α to normal cardiac function and immune regulation[2,9]. Therefore, a detailed understanding of how PP2A-B56α heterotrimer assembly and substrate recognition is regulated may have profound medical implications in several diseases. B56α interacts with up to 100 cellular target proteins (Supplementary Data 1 and [10,11]) among which, it is best known by its capacity to inhibit oncogenic activity of MYC[12,13]. The B56 proteins recognize their target proteins via a conserved binding pocket that binds to short linear motifs (SLIMs) with a LxxIxE consensus sequence in the target protein[10,14–16]. However, the mechanisms that inhibit PP2A-B56α are poorly understood. More, precisely, it is unknown whether the LxxIxE-binding pocket on B56α is subject to endogenous regulation mechanisms.

In cancers, different PP2A complexes are regulated by protein interactions with proteins such as CIP2A, PME-1, SET, or ENSA/ARPP19[17]. Of these, CIP2A is widely over-expressed in different human cancer types and drives tumour growth both in xenograft and transgenic mouse models[18–24]. Among the oncogenic CIP2A-regulated PP2A targets, the best-known is MYC[5,21,25]. However, recent identification of CIP2A as a central DNA damage response (DDR) protein essential for BRCA-mutated triple-negative breast cancer (TNBC) cells[24,26], indicates that CIP2A has broader role in cancer beyond MYC regulation. Accordingly, CIP2A was found to regulate S/T phosphorylation of about 100 target proteins implicated widely in different pathological and physiological processes[27]. CIP2A also broadly regulates cancer cell therapy responses, and its over-expression clinically correlates with relapse from kinase inhibitor therapies[27–30]. Collectively these data identify CIP2A as a very attractive cancer therapy target protein. However, our mechanistic understanding of structural determinants of CIP2A´s oncogenicity is still very rudimentary. The only thus far available structure of CIP2A spans the first 560 amino acids of the protein and reveals that CIP2A is an obligate homodimer interacting with B56 proteins[31].

Here we take a multidisciplinary approach to uncover in molecular detail how CIP2A inhibits PP2A- B56α revealing a unique hijack and mute mechanism involving both disassembly of the PP2A- B56α trimer and direct inhibition of LxxIxE motif recognition by B56α. The cancer relevance of the results is demonstrated by the abrogation of tumour growth of TNBC cells by single point mutation of the N-terminal B56α interaction domain of CIP2A. Thereby, beyond their general relevance to understanding phosphoregulation, the results can guide the development of therapies targeting CIP2A-B56α interaction in cancer, and other CIP2A-related diseases[2,32].

## Results

### Chemical cross-linking coupled to mass spectrometry shows that the CIP2A N-terminal head domain mediates B56α binding

The mechanism by which CIP2A inhibits PP2A-B56α is unknown (crystallization of CIP2A-B56α complex has not been successful[31] and unpublished data by Zhizhi Wang and Wenqing Xu). Therefore, we used chemical cross-linking coupled to mass spectrometry (XL-MS) to determine how CIP2A binds B56α. As a CIP2A protein we used the 1-560 fragment for which the crystal structure is available, and which is the longest fragment that can be purified in a soluble format from bacterial cultures[31]. DSS (disuccinmidyl suberate) and DMTMM (4-(4,6-dimethoxy-1,3,5-triazin-2-yl)−4-methyl-morpholinium chloride) were used to cross-link CIP2A-B56α complexes; DSS reacts with primary amines while DMTMM cross-links primary amino groups and negatively charged side chains (Supplementary Figs. S1A–S1C). The data show that the CIP2A(1-560) residues cross-linked to B56α are located primarily in the N-terminal head domain (K18, K21, E23, E34, K40) with an additional residue at the C-terminus (K490) (Fig. 1A and Supplementary Fig. 2). In B56α, two regions mediate CIP2A binding, an N-terminal region (region 1, residues 80-227), and a C-terminal region (region 2, residues 347-465) (Fig. 1B). While region 1 was expected based on previous yeast two hybrid (Y2H) experiments[31], region 2 has not previously been proposed to mediate CIP2A interaction. The selectivity of the observed cross-links is supported by the observation that although there are dozens of potentially reactive residues distributed throughout the CIP2A and B56α structures, only a very small subset was observed to form intermolecular cross-links (compare Fig. 1A, B to Supplementary Fig. 3). Also, the cross-linking results were essentially unchanged in the presence of increasing DSS cross-linker concentration (up to 500 μM; Figs. S2A, B), indicating the specificity of intra- and inter-molecular interactions. Finally, the comparison of the cross-linking results between CIP2A alone and the CIP2A:B56 complex showed that the frequency of CIP2A-specific intramolecular loops was reduced when CIP2A is in complex with B56α (Supplementary Fig. 2A, B), indicating that interaction with B56α locks CIP2A in stabilized structural conformation. Nevertheless, it is important to consider the XL-MS analysis as a screening experiment the results of which need further experimental validation and that the XL-MS data may contain some cross-links that are not functionally relevant.

Identification of the N-terminus of CIP2A as the primary B56α interaction motif is consistent with previous data demonstrating that CIP2A(1-330) fragment was sufficient for direct B56α interaction[31]. On the other hand, previously observed stabilization of CIP2A-B56α interaction by the C-terminal region 331-560[31] could be explained by the K490 cross-link (Fig. 1A). In order to further validate the individual contributions of the N- and C-terminal halves of the CIP2A(1-560) for B56α binding, we also subjected the CIP2A(1-330) fragment to cross-linking with B56α (Fig. 1C, D). When comparing the cross-linked CIP2A amino acids between CIP2A(1-330) and CIP2A(1-560), the CIP2A(1-330) recapitulated all N-terminal cross-links seen with CIP2A(1-560) (Fig. 1C). On the other hand, the number of CIP2A cross-links to B56α was clearly higher with CIP2A(1-560) than with CIP2A(1-330) (Fig. 1D) supporting the role of 331-560 region in mediating the interaction stability. The cross-linked amino acids on the N-terminal head domain of CIP2A comprised a ridge-like structure protruding from the rest of the head domain (Fig. 1E). Further, the shortest N-terminal CIP2A fragment that allowed to be bacterially expressed (aa. 1-85) was found to directly interact with B56α (Supplementary Fig. 4A). These data indicate that the head domain of CIP2A comprises an autonomous B56α interaction motif. Of note, the head domain of CIP2A was found to be evolutionary highly conserved in the animal kingdom from humans to the fish-like lancelet *Branchiostoma floridae* (Supplementary Fig. 4B).

Based on these results, we conclude that the N-terminal head domain of CIP2A(1-560) (Fig. 1E) is the primary region mediating B56α interaction, whereas the C-terminal region of CIP2A(1-560), involving K490, stabilizes the CIP2A-B56α interaction.

### N-terminal head domain of CIP2A stabilizes full-length CIP2A protein in cancer cells

To characterize the functional relevance of interaction between CIP2A head domain and B56α, we performed mutagenesis spanning the first 40 amino acid residues of CIP2A (Fig. 2A). The mutations were introduced to V5-tagged full-length CIP2A(1-905) mammalian expression

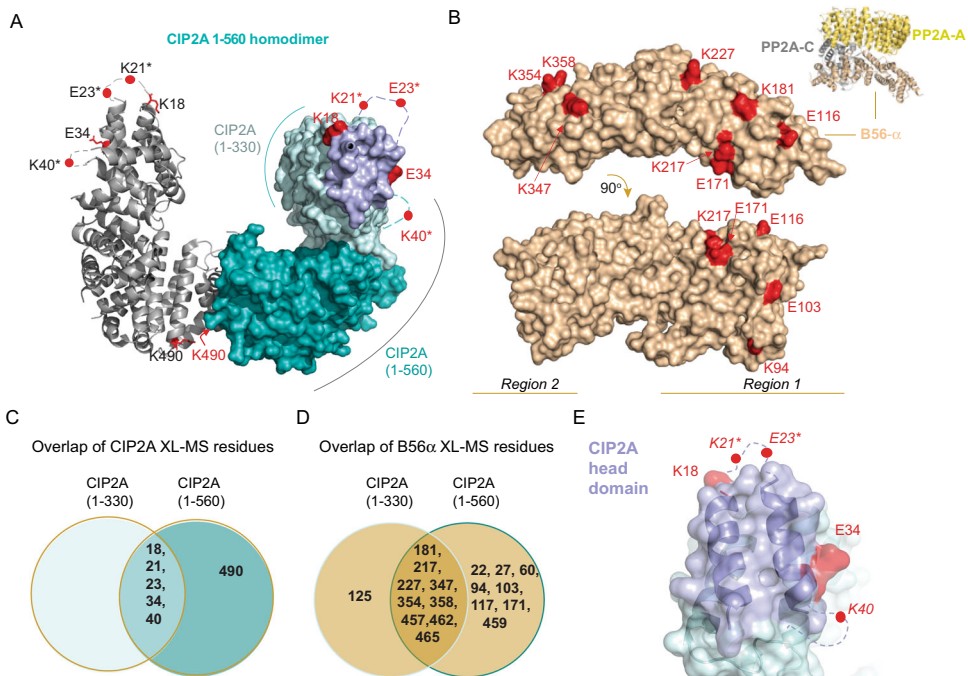

**Fig. 1 | CIP2A interacts with B56α via its N-terminal head domain. A** In red are indicated CIP2A(1-560) residues identified in inter-molecular cross-links to B56α using DSS and DMTMM cross-linkers. CIP2A 1-560 homodimer (PDB: 5UFL) monomers are shown either as ribbon or space filling models. In the space filling model 1-330 amino acids are in cyan including in purple the CIP2A head domain identified in this study. The 331-560 region including the dimerization domain is shown in turquoise. Due to high molecular flexibility of the N-terminal region of CIP2A, positions of residues K21 and E23 (indicated with*) had to be approximated using A24 position and K40 using position of R43. **B** Distribution of B56α (PDB: 6NTS) residues cross-linked to CIP2A by using DSS and DMTMM chemistries. Cross-

linked amino acids are in red and roughly distributed to two distinct regions on B56α shown in two different B56α orientations. Insert shows the overall structural organization of PP2A-B56α trimer consisting of PP2A-A, PP2Ac, and B56α. **C** Overlap of the B56α cross-links identified for two different CIP2A fragments, 1-330 and 1-560. **D** Overlap of the CIP2A cross-links on B56α using CIP2A fragments, 1-330 and 1-560. **D, E** The indicated cross-link sites are combined from both the DSS and DMTMM experiments. **E** Zoom-into the CIP2A´s head domain, shown for one CIP2A monomer. CIP2A sites found in cross-links, with DSS and DMTMM, to B56α are in red, and the head domain (1-43 aa) is coloured in purple. Residue annotation is the same as in panel (A).

constructs, and CIP2A protein expression was examined in 22RV1 prostate cancer cells. The chosen strategy was supported by previous data that CIP2A protein stability in cells directly reflects its B56 binding efficiency[31]. Furthermore, this was the only available approach to study the contribution of these N-terminal head domain amino acids in the context of full-length CIP2A protein because production of recombinant full-length CIP2A protein, in quantities and qualities required for functional experiments, remains elusive.

Notably, strongly indicative for the functional importance of the head domain, all six tested mutants showed an effect on full-length V5-CIP2A(1-905) protein levels, as compared to the wild-type protein (Fig. 2B, see Supplementary Fig. 5A for Western blots). In particular, the K8A single point mutation practically abolished, whereas K21A and K40A mutations significantly inhibited full-length CIP2A protein expression. On the other hand, A24E mutation increased full-length CIP2A protein expression when compared to the wild type (WT) (Fig. 2B and Supplementary Fig. 5A). Acknowledging the functional importance of high CIP2A protein expression in human cancers[18,21,23,24,26,33], we explored the potential impact of cancer-derived CIP2A head domain mutations on the protein expression. Based on positioning of the mutated residues and on the anticipated effect of mutation on B56α binding, we selected three CIP2A head mutations (Q16E, S22A, E23K) from the COSMIC and cBioPortal databases. Notably, when tested in MDA-MB-231 TNBC cells, dependent of CIP2A protein expression for their tumour growth[34], Q16E showed comparable CIP2A stabilization as with the A24E mutant when compared to the V5-tagged WT protein (Fig. 2C).

Corroborating with the decreased full-length CIP2A protein expression in 22RV1 cells (Fig. 2B), bacterially produced CIP2A(1-560) K8A mutant demonstrated decreased direct B56α binding as

compared to the WT CIP2A(1-560) in in vitro pull-down assay (Fig. 2D, E). Also the recombinant CIP2A(1-560) K21A mutant was found to interact less efficiently with B56 than the CIP2A 1-560 WT protein (Supplementary Fig. 5B, C). The A24E mutation, when tested in the context of the N-terminal CIP2A(1-330) fragment, instead showed increased direct B56α binding (Fig. 2F, G). Due to their instability, the other tested head domain mutations could not be expressed as bacterial proteins. As these results strongly indicate that effects of K8A and A24E mutations on CIP2A protein stability is linked to their B56α-binding propensity, we asked whether the decreased cellular stability of K8A mutant full-length CIP2A could be rescued by increasing its B56α binding affinity via concomitant A24E mutation. Indeed, when MDA-MB-231 cells were transiently transfected with a construct carrying both A24E and K8A mutations, we observed that full-length CIP2A protein expression was rescued to the WT level (Fig. 2H, I). As a molecular explanation for the CIP2A protein stability changes, we examined potential ubiquitination sites on the full-length protein. Previously CIP2A protein stability has been shown to be regulated by CHIP-mediated ubiquitination[35], but the ubiquitination site on CIP2A has not been characterized. Based on database information (https://www.phosphosite.org/), lysine 647 (K647) is the most prevalently ubiquitinated CIP2A amino acid. Notably, K647A mutation rescued the expression of CIP2A K8A mutant in MDA-MB-231 cells, but did not further increase the expression of already stabilized A24E mutant (Supplementary Fig. 5D, E).

Collectively, these data demonstrate that the N-terminal head domain of CIP2A mediating interaction with B56α is a critical determinant of full-length CIP2A protein expression in cancer cells. The results further indicate that decreased stability of N-terminal head mutants is mediated by C-terminal K647, which may be differentially

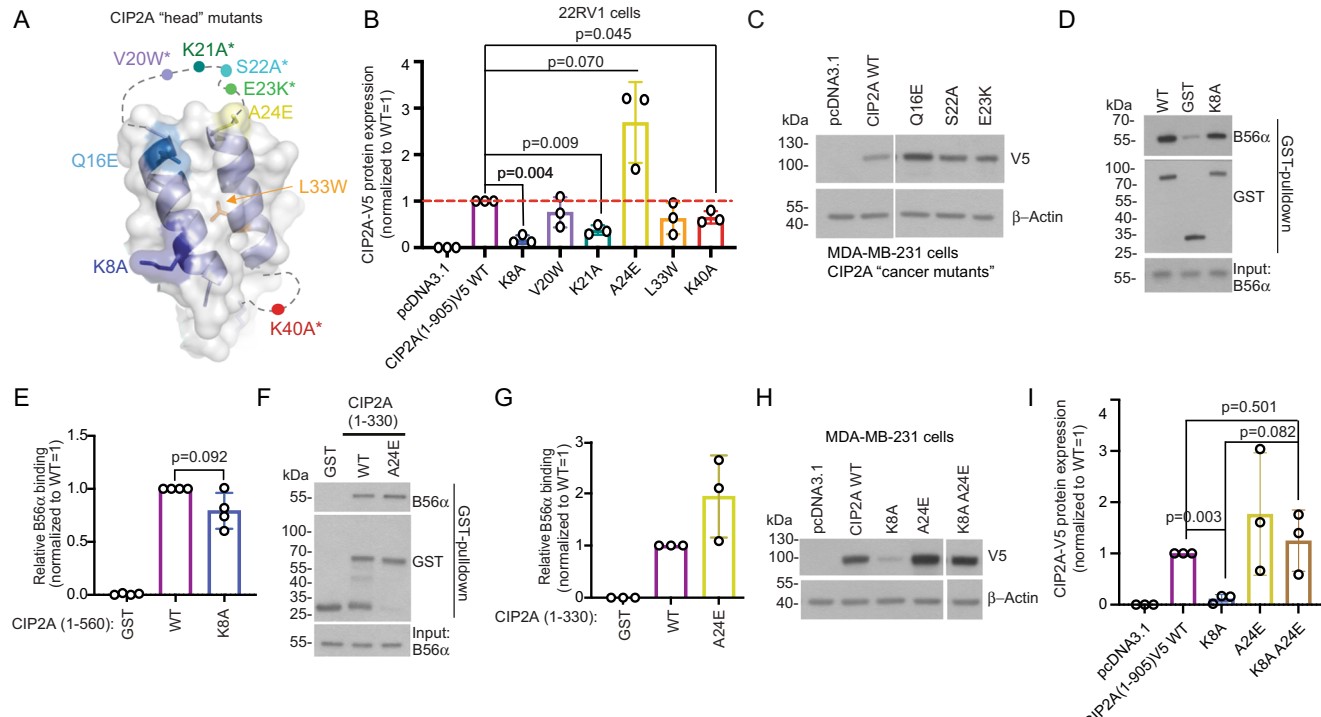

**Fig. 2 | The N-terminal head domain stabilizes CIP2A protein expression in cancer cells. A** Mutations on the N-terminal head of CIP2A (PDB: 5UFL) are indicated with different colours used also in bar graphs B,E,G,I. **B** Transient overexpression of full-length CIP2A(1-905)V5 variants in prostate cancer 22RV1 cells. Representative Western blot images are shown in S5. Quantification is shown as a mean + S.E.M. from $N = 3$ biological repeats. Expression levels of CIP2A mutants were normalized to expression of WT CIP2A, which was set as value 1 in each experiment. Two-tailed One sample Wilcoxon t test. Source data are provided as a Source Data file. **C** Impact of cancer-derived CIP2A head domain mutants on CIP2A full length protein expression in breast cancer MDA-MB-231 cells. **D, F** GST pulldown assay comparing direct B56α interaction between recombinant GST-tagged CIP2A(1-560) WT protein and K8A or A24E mutants. Source data are provided as a

Source Data file. **E, G** Quantification of the data from D and F. B56α signal from eluates were normalized to GST signal. Quantification is shown as a mean + S.E.M. from $N = 4$ and 3 biological repeats, respectively. B56α signal from mutants were normalized to signal from WT CIP2A, set as a value 1 in each experiment. **E** Two-tailed One sample Wilcoxon t test. **H** Analysis of combinatorial effects of stabilizing and destabilizing full-length CIP2A(1-905)V5 head mutants by transient overexpression in triple negative breast cancer MDA-MB-231 cells. Source data are provided as a Source Data file. **I** Quantification of (**H**). Shown is a mean + S.E.M. from $N = 3$ biological repeats. Expression levels of CIP2A mutants were normalized to expression of WT CIP2A set as value 1 in each experiment. Two-tailed One sample Wilcoxon t test.

exposed to ubiquitination depending on the tightness of CIP2A-B56α interaction.

## Single amino acid mutation on the N-terminal head domain of CIP2A abrogates TNBC tumorigenicity

To test the cancer relevance of the N-terminal head domain of CIP2A, we created MDA-MB-231 single cell clones carrying K21A mutation by CRISPR/Cas9. K21A mutation was chosen due to its intermediate destabilizing effects on CIP2A protein expression (Fig. 2B), as the near complete loss of CIP2A protein expression as observed with K8A mutant, was expected to prevent growth of single cell MDA-MB-231 clones based on previous CRISPR/Cas9 knock-out data (Avana 2020 Q1; https://depmap.org)[24]. K21 was targeted by two independent guide RNAs (gRNA) using direct ribonucleoprotein (RNP)-mediated transfection[36]. Both gRNAs resulted in genomic CIP2A targeting (Supplementary Fig. 6A) and in decreased CIP2A protein expression in transfected cell pools (Supplementary Fig. 6B). Based on more robust impact on CIP2A expression, gRNA #2 cell pool was selected for single cell cloning. Out of sequenced single cell clones, we could verify K21A mutation from two clones (clone1 and clone2) (Supplementary Fig. 6C) which were in experiments compared to the control cells that were RNP-transfected without gRNA.

Decreased expression of CIP2A protein in both single cell clones was verified by Western blotting (Fig. 3A). CIP2A targeting also inhibited expression of validated PP2A-B56α target MYC (Fig. 3A). Importantly, when assessed by proximity ligation analysis (PLA), the K21A

mutation on the full-length endogenous CIP2A protein almost entirely abolished the CIP2A-B56α interaction (Fig. 3B, C), although there was still approximately 50% of K21A mutant protein expressed in the cells used for this assay (Supplementary Fig. 6D). The specificity of the PLA reaction was tested by performing the assay without CIP2A primary antibody (Supplementary Fig. 6E). These data further validate critical role of the N-terminal head domain on the full-length CIP2A interaction with B56, and on CIP2A protein stability in cellulo.

Interestingly, the colony formation potential in plastic was comparable between the two CIP2A hypomorph clones and the control cells (Fig. 3D, E). However, fully consistent with the requirement of PP2A-B56 inhibition for anchorage-independent growth[6,7], CIP2A K21A targeted clones were almost completely incompetent to grow as spheres on soft agar (Fig. 3F, G). The data above demonstrated decreased interaction between K21A mutant of CIP2A and B56 in vitro and in cellulo. However, it was yet unclear how much the decreased protein expression of K21A mutant of CIP2A contributed to loss of malignant growth on soft agar. To address this, we created two independent CIP2A shRNA clones that have approximately similar levels of CIP2A protein expression than was observed in K21A mutant Crispr clones (Supplementary Fig. 6F). We also titrated CIP2A siRNA to the levels that only partially inhibited CIP2A expression (Supplementary Fig. 6G). Notably, the both types of CIP2A hypomorph MDA-MB-231 cells were yet fully capable in forming colonies in soft agar (Supplementary Fig. 6H–K). This demonstrates that the total loss of malignant growth potential of K21A mutant Crispr clones (Fig. 3F, G) cannot be

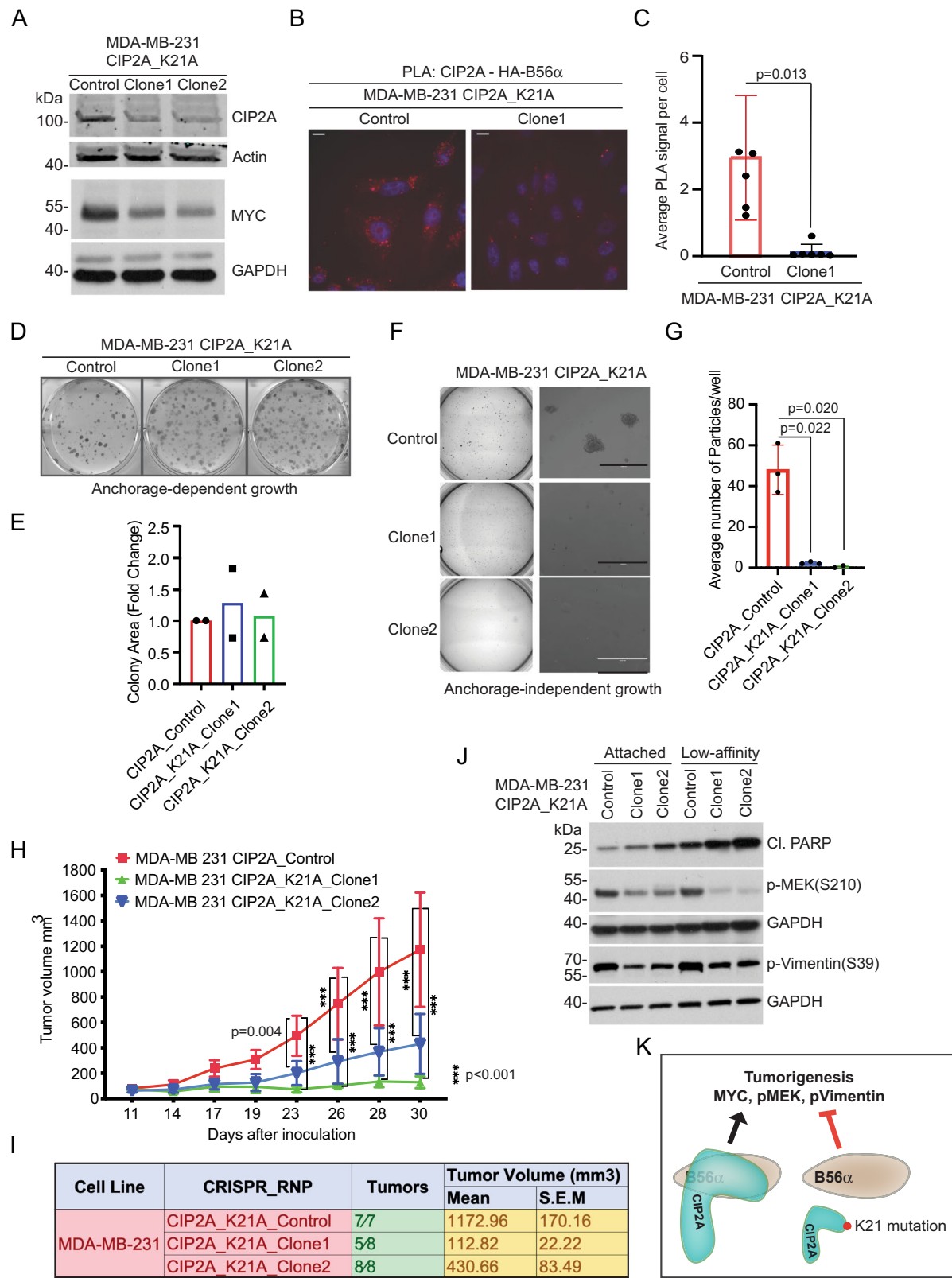

explained only by partial inhibition of CIP2A protein expression in these clones, but it also involves loss of CIP2A-B56 interaction.

Based on the results that single K21 amino acid on CIP2A is imperative for transformed cell growth, the effects of K21A mutation on MDA-MB-231 cell tumorigenicity was tested by orthotopic xenograft assay. Notably, both clones showed significantly impaired tumour growth as compared to control cells (Fig. 3H). Especially with

clone 1, only 5/8 tumours were palpable, and the tumours grew very slowly reaching the average size of only 113 mm³ after 30 days (Fig. 3I). As CIP2A K21 was selectively essential for anchorage-independent growth of MDA-MB-231 cells in vitro (Fig. 3F, G vs. D, E) and in vivo (Fig. 3H, I), we assessed both induction of anoikis (apoptosis due to loss of cell adhesion), and PP2A-regulated phosphotargets related to cell survival in anchorage-independent conditions. To this end, PARP

**Fig. 3 | Single amino acid mutation on the N-terminal head domain of CIP2A abrogates TNBC tumorigenicity. A** Western blot analysis of CIP2A and MYC protein expression in two independent MDA-MB-231 CIP2A K21A mutant clones. Source data are provided as a Source Data file. **B** Proximity ligation assay (PLA) for interaction between HA-B56α and endogenous CIP2A. MDA-MB-231-Control and MDA-MB-231_K21A mutant clone1 cells transfected with HA-B56α construct were subjected to PLA with anti-HA and anti-CIP2A antibodies. Red dot indicates the association between HA-B56α and endogenous CIP2A proteins. Shown is a representative image from $N = 3$ PLA experiments. Scale bar = 10 μm. **C** Quantification of PLA shown in (**B**) using automated macro detecting PLA signals. Error bar represent mean +/− SD from 3 different fields from 2 technical replicates. Two-sided unpaired t-test with Welsh correction. **D** Anchorage-dependent colony growth potential of MDA-MB-231 control and CIP2A K21A mutant clones. **E** Quantification of (**D**). Shown is mean of two technical replicates from each indicated cell line ($n = 2$).

**F** Anchorage-independent colony growth potential of MDA-MB-231 control and CIP2A K21A mutant clones on soft agar. **G** Quantification of (**F**). Shown in mean +/− S.E.M of three technical replicates from each indicated cell line. Two-tailed t-test with Welsh correction. **H** Growth curves of orthotopic mammary fat pad xenograft tumours from MDA-MB-231 control ($n = 7$ mice) or CIP2A K21A mutant clones ($n = 8$ mice). Error bar represent mean +/− S.E.M. Two-sided Anova. **I** Numbers of detectable tumours and their mean volumes at day 30 from (**H**). **J** Western blot analysis of cleaved PARP (Cl.PARP), phosphorylated MEK and phosphorylated Vimentin from two independent MDA-MB-231 CIP2A K21A mutant clones cultured either on normal plastic or on low-attachment polyHEMA coated plates. Source data are provided as a Source Data file. **K** Schematic presentation how K21A mutation on CIP2A leads to protein destabilization and thereby releases B56 from CIP2A-mediated inhibition. This results in dephosphorylation of PP2A-B56 targets and in inhibition of tumorigenesis.

cleavage and phosphorylation of MEK MAPKK and vimentin were studied from clones cultured for 24 h on either plastic or on low-affinity polyHEMA-coated plates. Increased PARP cleavage confirmed the increased sensitivity or K21 targeted clones to anoikis induction under low-affinity conditions (Fig. 3J). Interestingly, phosphorylation of pro-survival MEK MAPKK was decreased in K21 targeted clones in both conditions, but the effect was more pronounced in cells from low-affinity plates providing a plausible explanation for increased anoikis[37] (Fig. 3J). Further, phosphorylation on PP2A target vimentin[38], supporting anchorage-independent growth[39], was strongly reduced in both K21A clones in both conditions.

Collectively, these results validate the cancer relevance of the N-terminal head domain of CIP2A. They also identify targeting of the head domain as an efficient strategy to block malignant growth of CIP2A-driven TNBC cells[24] (Fig. 3K).

## CIP2A hijacks B56α and PP2Ac from the PP2A-B56α heterotrimer

Although the results above clearly demonstrate the relevance of the N-terminal head domain on oncogenicity by CIP2A, it is still totally unknown how CIP2A mechanistically affects PP2A-B56α holoenzyme function. To address this, we mapped the CIP2A(1-560) cross-link sites to B56α structure in the PP2A-B56α trimer configuration. Interestingly, comparison of the hypothetical surface area which CIP2A(1-560) could cover on B56α based on the cross-link sites (Fig. 4A, red), to the known sites involved in B56α interaction with the A-subunit (Fig. 4A, purple)[40,41], revealed that these surface areas are highly overlapping (Fig. 4A, right panel, shaded circle). Also, one of the CIP2A cross-linking sites, K358, is adjacent to the PP2Ac interaction surface on B56α (Fig. 4A, right panel). These observations provoked us to hypothesize that CIP2A(1-560) binding to B56α might interfere with PP2A-B56α holoenzyme assembly.

To study this intriguing possibility, we incubated GST-fused CIP2A(1-560) with the pre-assembled PP2A-B56α holoenzyme (A-B56α-PP2Ac) in vitro and analyzed reconstitution of protein interactions from glutathione-bead immunoprecipitates by Coomassie gel stain. Supporting the structure-based hypothesis above, CIP2A was found to capture B56α and PP2Ac, but not the scaffolding A subunit from the PP2A trimer (Fig. 4B). These results were confirmed by Western blot analysis of the glutathione-bead immunoprecipitates from independent experiments (Fig. 4C). Consistently with other data, no direct interaction between CIP2A(1-560) and A-subunit could be detected in a GST pull-down experiment (Supplementary Fig. 7A). These results were substantiated by demonstration of CIP2A concentration-dependent displacement of the scaffold subunit, and extraction of B56 and PP2A-c from the holoenzyme (Fig. 4D and Supplementary Fig. 7B). Importantly, the data also demonstrated higher capability of CIP2A(1-560) to extract B56α from the holoenzyme than CIP2A(1-330) (Fig. 4D). This is consistent with the role of C-terminal interaction region in stabilizing the CIP2A-B56 interaction[31] (Fig. 1C, D).

Interestingly, while CIP2A-PP2Ac interaction has not been previously reported, their direct interaction was confirmed by GST pull-down experiment (Supplementary Fig. 7C).

The results above imply a model where CIP2A can potentially function as a pseudo A-subunit for a trimeric CIP2A-B56α-PP2Ac complex. To demonstrate that such a CIP2A-B56α-PP2Ac complex can be reconstituted, we incubated the pre-assembled PP2A holoenzyme (A-B56α-PP2Ac) with either GST only, or with GST-CIP2A(1-560), and subjected the samples for analytical gel filtration analysis (Fig. 4E, F and Supplementary Fig. 7B). Reassuringly, in the control samples co-incubated with GST (Fig. 4E, middle panel), the A-B56α-PP2Ac trimer components co-eluted in fractions 3 and 4, and without GST, whereas in the samples incubated with GST-CIP2A(1-560) (Fig. 4F, middle panel), CIP2A co-eluted with B56α and PP2Ac without the A-subunit in the fractions 2-4. We also conducted gel filtration to mimic conditions that would exist following exclusion of A-subunit from the CIP2A-B56α-PP2Ac trimer. Therefore we incubated either GST alone, or GST-CIP2A(1-560), with B56α and PP2Ac, which do not constitute a stable dimer without any scaffolding protein[40,41]. Consistent with this previous data, B56α and PP2Ac did not co-elute when incubated with GST (Fig. 4E, lower panel). However, in the presence of GST-CIP2A(1-560), PP2Ac and B56α co-eluted in high molecular weight complexes 2 and 3 containing also GST-CIP2A(1-560) (Fig. 4F, lower panel).

These results reveal reorganization of PP2A-B56α holoenzyme upon CIP2A binding, resulting in formation of unforeseen trimeric CIP2A-B56α-PP2Ac complex (Fig. 4G).

## CIP2A mutes LxxIxE motif-dependent substrate binding to B56α

The results above identify a mechanism by which CIP2A hijacks B56α and PP2Ac from the PP2A-B56α complex. However, if CIP2A would merely function as a pseudo-A subunit, it is unclear whether the newly formed CIP2A-B56α-PP2Ac heterotrimer could still recognize B56α targets. Excitingly, we made a notion that the CIP2A(1-560) cross-linking sites on B56α (aa. 171, 181, 217, and 227), were lining the groove which recognizes the LxxIxE motif in B56α substrates (Fig. 5A). On the other hand, the recently identified additional determinant for high affinity binding of LxxIxE substrates to B56α, an acidic patch between B56α amino acids E301-E341[42], is sandwiched between the LxxIxE motif, and the C-terminal CIP2A cross-link sites in B56α (aa. 347, 354, and 358) (Supplementary Fig. 8A). These results indicate that CIP2A could mute B56α in the CIP2A-B56α-PP2Ac trimer by sterically preventing recognition of the LxxIxE motif containing substrate proteins. To study this possibility, we used a LxxIxE motif-containing protein fragment from BubR1, which is the prototypic B56α substrate[15,43,44]. In the experiment, GST-tagged BubR1(647-720) was pre-incubated with B56α prior to the addition of V5-tagged recombinant CIP2A(1-560), and binding efficiency of B56α to BubR1 was assessed from the glutathione bead pull-down samples. Notably, whereas CIP2A was found not to interact with BubR1 (Fig. 5B, lane 2), it efficiently out-competed B56α from BubR1 (Fig. 5B, C, lane 1 vs lane 3).

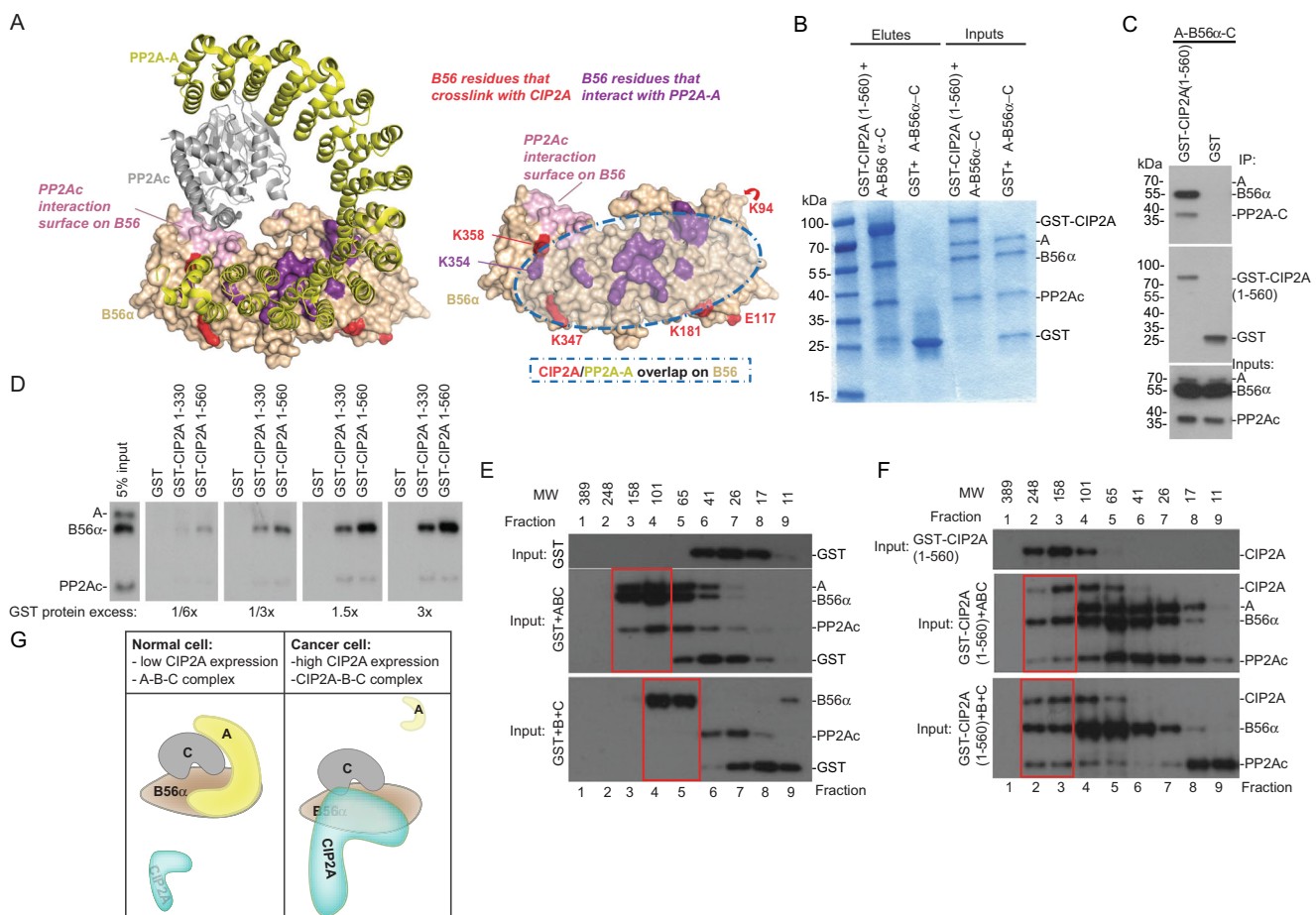

**Fig. 4 | CIP2A displaces PP2A-A from PP2A-B56α trimer and interacts directly with PP2Ac. A** B56α sites identified in inter-molecular cross-links with CIP2A overlap with B56α contact sites with the scaffolding A subunit (PDB: 6NTS). A subunit is yellow, catalytic subunit is grey, and B56α is wheat. The XL-MS cross-links between CIP2A and B56α are in red. PP2A-A and PP2Ac interaction sites with B56α are shown in magenta, and light purple, respectively (based on refs. [40,41]). The overlap between PP2A-A and CIP2A in B56α surface is indicated as transparent oval shape in the right panel. **B** Coomassie stained SDS-PAGE of PP2A trimer components interacting with CIP2A(1-560) after incubation with the pre-assembled PP2A-B56α heterotrimer (A-B56α-PP2Ac). A = PP2A-A; C = PP2Ac. Representative image from four experiments is shown. **C** Western blot analysis of the similar experiment as in (**B**). The different intensities of the Western blot signals between PP2A-A (**A**), B56α, and PP2Ac in the input (and between B56α, and PP2Ac in the eluates), is due to differential affinities of the antibodies used simultaneously to blot the membranes. *N* = 3 biological repeats. **D** GST pull-down assay for PP2A trimer-GST-

CIP2A(1-560) interaction. Equal molar amounts of PP2A proteins were used in all the samples. The amount of CIP2A(1-560) protein was titrated against PP2A as indicated. The positions of molecular weight markers are unavailable. **E, F** Size-exclusion chromatography analysis of protein complexes between GST (**F**) or GST-CIP2A(1-560) (**G**) and either pre-assembled PP2A-A-B56α-PP2Ac trimer (ABC) (middle panels) or B56α and PP2Ac (B + C) proteins (lower panels). Proteins eluting from the indicated fractions were analyzed by Western blotting. Approximate molecular weights of protein complexes eluting from each fraction are based on calibration with standard proteins. The positions of molecular weight markers are unavailable. *N* = 1. **G** Schematic presentation of CIP2A mediated hijack of the PP2A-B56α complex. In normal cells with low CIP2A expression, the PP2A-B56α holoenzyme (A-B56α-C) remains intact. In cancer with high CIP2A expression CIP2A interacts directly with both B56α and PP2Ac resulting in expel of A-subunit from the trimer. Thus, in cancer cells CIP2A functions as a pseudo-A-subunit stabilizing the trimeric CIP2A-B56α-PP2Ac complex.

To directly link these results to structural requirements of CIP2A-B56α interaction, we tested whether the head domain peptide of CIP2A (aa.18-40) could alone block B56α-BubR1 interaction. In the experiment, B56α was pre-incubated with the CIP2A head peptide, after which the complexes were co-immunoprecipated with GST-BubR1. Fully consistent with our model, BubR1 binding to B56α was reduced in the presence of CIP2A head peptide (Fig. 5D, E). We further compared CIP2A binding to the B56α variants with groove mutations that abolish binding of the LxxIxE motif targets[10]. Notably, whereas both of the B56α variants tested, Y215Q and R222E, exhibit total loss of BubR1 binding[10], they had only slight impact on CIP2A binding: Y215Q did not influence CIP2A binding at all, and R222E had about 40% inhibitory effect as compared to B56α wild-type (Fig. 5F, G). These results demonstrate that CIP2A inhibits B56α binding to LxxIxE motif substrate, but is itself not a classical LxxIxE motif substrate.

To address whether the cross-linked residues neighbouring the LxxIxE binding groove on B56 have a role on either CIP2A binding, or on PP2A holoenzyme assembly, we mutated the K181, K217 and K227 to alanines. In in vitro GST pull-down assay, these mutations did not impact direct CIP2A-B56 interaction (Supplementary Fig. 8B). This could be explained by multiple other CIP2A-B56 cross-link sites potentially stabilizing the interaction enough under these experimental conditions. However, when tested *in* cellulo, these B56 mutations had an instrumental role in stability of PP2A trimer, as demonstrated by very potent inhibition of both B56-A and B56-PP2Ac interaction, and loss of catalytic PP2A activity (Fig. 5H and Supplementary Fig. 8C) from the B56 mutant protein pull-down samples. These results demonstrate that in addition to its critical role in substrate recognition, the LxxIxE groove region on B56 has also an important role in PP2A trimer stability. This is best explained by polar

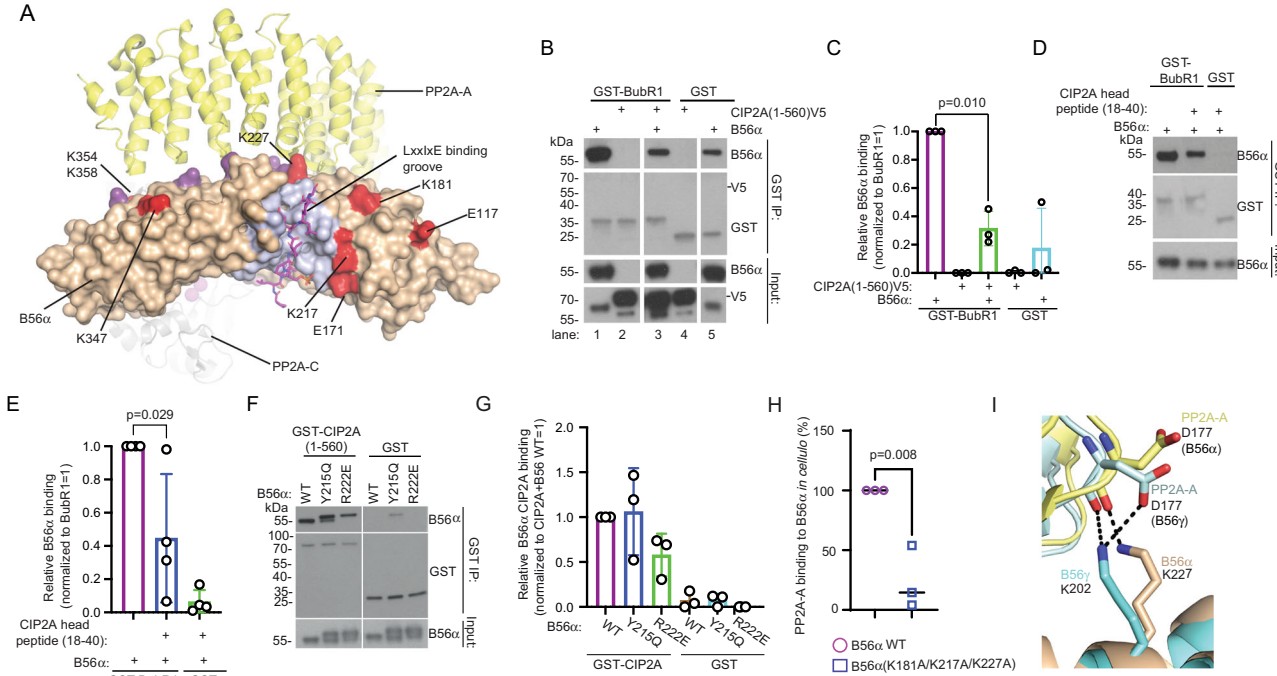

**Fig. 5 | CIP2A mutes LxxIxE-motif substrate binding of B56α. A** Sites of CIP2A-B56α cross-links (in red; PDB: 6NTS) mapped in relation to indicated LxxIxE-binding groove of B56α. Residues indicated in light purple constitutes the LxxIxE-binding region of B56α between amino acids 212-271. Residues in dark purple indicate amino acids involved in PP2A-A interaction as explained in Fig. 4. **B** CIP2A(1-560)V5 out-competes prototypical LxxIxE motif target BubR1(LxxIxE peptide 647-720) from its direct association with B56α. Source data are provided as a Source Data file. **C** Quantification of GST pull-down data from (**B**) shown as a mean + S.E.M from $N = 3$ biological repeats. Two-sided t-test. **D** B56α competition assay using GST-BubR1(647-720) alone, or in combination with CIP2A N-terminal head peptide (aa. 18-40). Source data are provided as a Source Data file. **E** Quantification for data from D shown as a mean + S.E.M from $N = 4$ biological repeats. Two-sided t-test. **F** In vitro binding assay using purified recombinant GST-tagged CIP2A(1-560) and B56α WT or B56 variants Y215Q and R222E, which contain mutations that prevent interaction with LxxIxE groove substrate proteins[10]. Source data are provided as a Source Data file. **G** Quantification of data from (**F**) shown as mean + S.E.M from $N = 3$ biological repeats. **H** GFP-tagged B56α WT or indicated triple mutant were expressed in HEK-293-T cells. The amount of PP2A-A subunit bound to GFP-tagged B56α upon GFP trap pull-down was quantified by anti-PP2A-A immunoblotting. Shown are the mean values + S.E.M of the ratios of the quantified anti-PP2A-A signal versus the quantified anti-GFP signal, relative to the B56α WT (set at 100 %) from $N = 3$ biological replicates. A two-sided one-sample t-test. **I** Triple B56α mutant (K181A/K217A/K227A) exhibits loss of K227 (B56)-D117(PP2A-A) salt bridge. Overlay of PP2A-B56α (PDBID 6NTS: PP2A-A, yellow; B56α, beige) and PP2A-B56γ (PDBID 2IAE: PP2A-a, light cyan, B56γ, cyan), with PP2A-A D177 and B56α/B56γ K227/K202 residues shown as sticks and labelled. H-bond interactions are shown using dotted lines. Image was generated using Pymol.

interactions between D177 of the PP2A-A subunit and K227 of the B56α (Fig. 5I; PDBID 6NTS). Consistent with the conserved interaction between CIP2A and both B56α and B56γ[31] (Supplementary Fig. 5B, C), D177 of PP2A-A interacts in a similar fashion with K202 of B56γ (Fig. 5I; PDB: 2IAE[41]).

These results provide mechanistic explanation for displacement of the PP2A-A subunit from B56 upon CIP2A binding to the LxxIxE region of B56.

### Validation of the "hijack and mute" model of PP2A-B56α inhibition by CIP2A *in cellulo*

The above in vitro results reveal an unprecedented hijack and mute mechanism of PP2A holoenzyme inhibition. In this model, CIP2A both expels the A-subunit from the PP2A-B56α trimer and subsequently inhibits B56α binding to LxxIxE motif substrates. To validate these two inhibitory modes *in cellulo*, we first analyzed the impact of CIP2A on B56α interactome by affinity purification coupled to MS (AP-MS) approach[45]. For that purpose, we used murine NIH3T3 cells stably expressing either human CIP2A(1-905)V5, or empty vector control (Fig. 6A and Supplementary Fig. 8D). NIH3T3 cells do not express endogenous CIP2A protein, but CIP2A over-expression increases their proliferation potential[21]. These features make them a perfect cellular model for studying PP2A-B56α inhibition resulting from CIP2A over-expression. These stable cells were transiently co-transfected with YFP-tagged B56α, and GFP-Trap immunoprecipitation followed by MS

analysis was used to identify B56α interacting complexes (Fig. 6A). The results of AP-MS experiment with three replicate samples confirmed association of CIP2A with B56α in CIP2A(1-905)V5 overexpressing cells only (Fig. 6B and Supplementary Data 2). Further, fully supportive of the model that CIP2A can expel A-subunit from the A-B56α-PP2Ac trimer, A-subunit (PPP2R1A) association with B56α was significantly decreased in CIP2A over-expressing cells (Fig. 6B). On the other hand, validating the function for CIP2A in preventing B56α-substrate inter-actions, among all B56α co-precipitating proteins for which CIP2A over-expression had a significant impact (> Log2 fold change 1, $p < 0.05$), 80% of proteins showed decreased B56α binding (Fig. 6B, C). Notably, 63% of these proteins had a candidate LxxIxE motif (Fig. 6C and Supplementary Data 2) which is consistent with our in vitro results.

To substantiate these conclusions, we validated the impact of N-terminal K21A mutation on phosphorylation of LxxIxE-motif containing proteins. Comparison of LC-MS analyzed phosphoproteomes between the control clone and CIP2A K21A clone 1 (both in triplicates) resulted in identification of 78 phosphopeptides from 63 individual proteins that were significantly (FDR < 0.05) downregulated in CIP2A K21A cells (Supplementary Data 3). The entire MS/MS identification data can be found from supplementary Data 4 and from GEO database (PXD035179).

Notably, 53% of the proteins dephosphorylated CIP2A K21A cells had a candidate LxxIxE motif (Fig. 6D and Supplementary Data 4). It was previously shown that LxxIxE containing proteins can act as

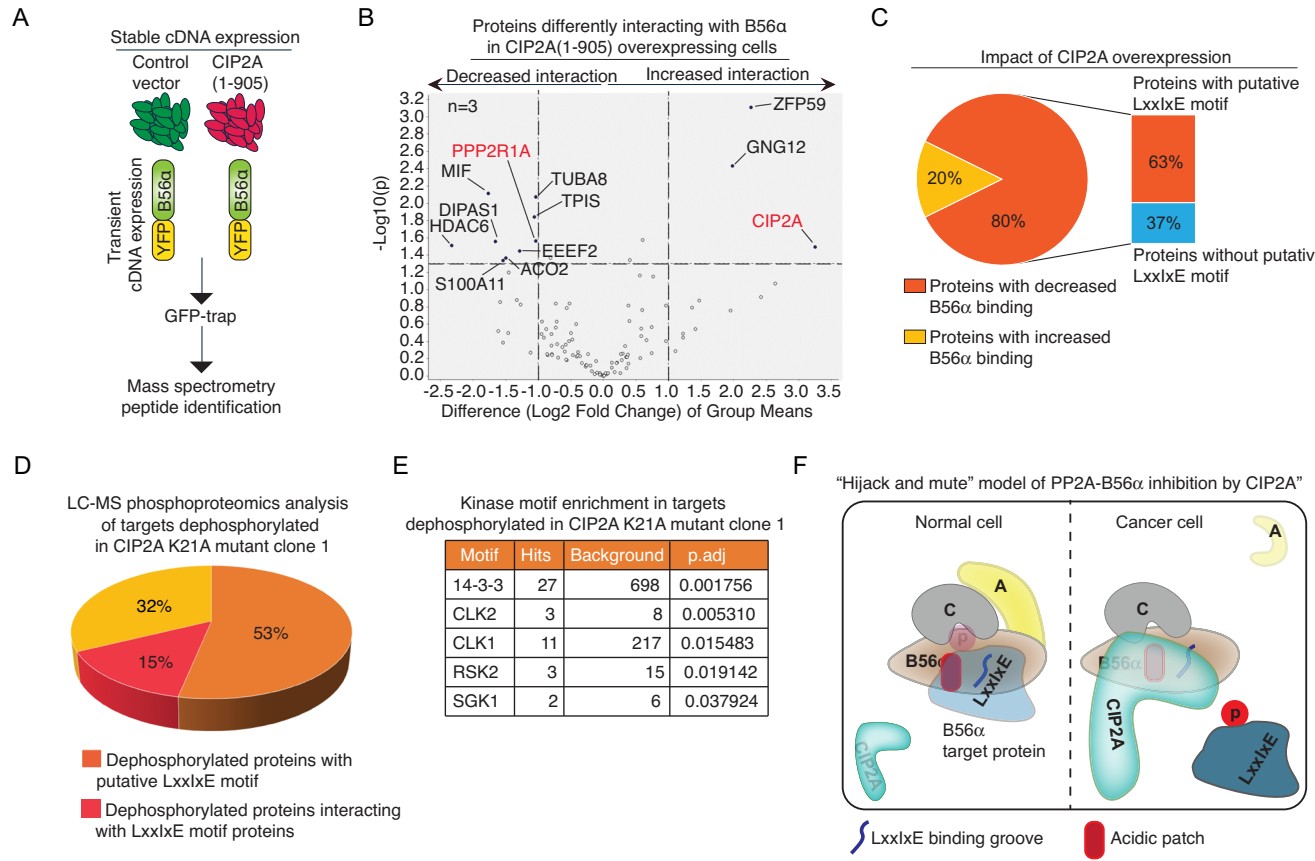

**Fig. 6 | Validation of hijack and mute model of B56α inhibition by CIP2A *in cellulo*. A** NIH3T3 cells, stably expressing human CIP2A(1-905) V5, or empty vector control (see Supplementary Fig. 8D), were transiently transfected with YFP-B56α WT, followed by immunoprecipitation using GFPTrap and analysis by mass spectrometry. **B** Proteins differentially interacting with B56α in CIP2A(1-905)V5 over-expressing cells (*N* = 3) vs. control (*N* = 3). Indicated proteins exceed the threshold of 2-fold difference with *p* < 0.05 (two-sided *t*-test using log2 of protein abundances, normalized to the protein abundance of B56α in the CTRL_1 sample, as an input). **C** Pie-chart of share of proteins displaying either decreased or increased binding to B56α upon CIP2A over-expression (excluding PPP2R1A and CIP2A). Bar graph indicates percentual share of proteins with putative LxxIxE motif among the proteins which interaction with B56α was decreased by CIP2A. **D** Pie-chart presentation of percentual share of proteins dephosphorylated (FDR < 0.05) in CIP2A K21A clone1 as compared to control clone in relation to putative LxxIxE motif found from the dephosphorylated protein (in orange), or whether dephosphorylated

protein is an interactor of a protein with validated LxxIxE motif (in red). **E** Significantly enriched kinase motifs based on phosphopeptides dephosphorylated in CIP2A K21A clone1. Background indicates the number of motifs found from all phosphopeptides identified by MS from the same samples. **F** Reactome analysis image capture of the process Senescence enriched in targets dephosphorylated in CIP2A K21A clone1 as compared to control clone. Schematic presentation of CIP2A-mediated PP2A-B56α inhibition by the identified hijack and mute mechanism. In normal cells with low CIP2A expression, PP2A-B56α binds to its substrate protein (light blue) via LxxIxE groove and the adjacent acidic patch (orange) resulting in substrate dephosphorylation. In cancer cells with high *CIP2A* gene transcription, CIP2A binding to PP2A-B56α trimer results in expel of A and the formation of CIP2A-B56α-PP2Ac trimer. In this alternative trimer CIP2A shields the LxxIxE groove and the adjacent acidic patch from B56α substrates and thereby the substrate remains phosphorylated (dark blue). Binding of CIP2A to B56α via its head domain further stabilizes CIP2A protein.

---

scaffolds for the recruitment of other proteins for PP2A-B56α mediated dephosphorylation[46]. Using the PINA.3 database[47], we found that an additional 15% of proteins identified in our screen are interactors of proteins with validated or predicted LxxIxE docking motifs[46,48] (Fig. 6D and Supplementary Data 4). Thus, the 68% of proteins dephosphorylated in CIP2A K21A clones are either LxxIxE containing proteins, or their interactors, strongly supporting the notion that CIP2A inhibits substrate recognition of PP2A-B56. We also analyzed enrichment of kinase target motifs among the dephosphorylated peptides from CIP2A K21A cells using all quantified phospho-peptides from the same MS analysis as the background (Fig. 6E). Notably, among the enriched motifs, 14-3-3 protein binding motif is a firmly established PP2A-B56α target in many signal transduction proteins[49]. Further, recent studies indicate functional relationship between PP2A-B56 and CLK2 in different physiological and pathological context[50,51]. Lastly, it was recently reported that RSK2-mediated phosphorylation of Emi2 promotes its well-known interaction with PP2A-B56[52,53]. Finally, in a pathway analysis, CIP2A-regulated phosphoproteins were enriched

among senescence-associated proteins (Fig. 6F and Supplementary Data 5), which with the previous evidence linking CIP2A to senescence evasion[18], provides a plausible mechanistic explanation for the potent tumour growth inhibition in CIP2A K21A mutant clones.

Collectively, these results provide comprehensive *in cellulo* validation that CIP2A inhibits PP2A-B56α function by hijack and mute mechanism: by expelling the A-subunit from the trimer, and by shielding B56α from binding to its LxxIxE motif substrate proteins (Fig. 6F).

## Discussion

During the past two decades, there has been a major advancement in understanding how protein serine/threonine-specific protein phosphatases (PPPs), historically considered promiscuous enzymes lacking substrate specificity, can selectively dephosphorylate their substrates[1,3]. The key to the precision of most PPPs in substrate recognition is formation of dimeric, or trimeric, complexes between the catalytic subunit and different regulatory subunits; such as B-subunits for the trimeric PP2A complexes. Thereby, and especially as

different B-subunit containing PP2A complexes can have even opposite cellular functions, such as tumour suppression or oncogenesis[54], understanding the function and regulation of individual PP2A B-subunits is essential. Amino acid sequence determinants for selective recognition of target proteins have recently been discovered for some PPP regulatory subunits[1,10,14,42,49,55]. However, whether the regions of regulatory subunits mediating the substrate interactions. are themselves subject to regulation by endogenous mechanisms, is poorly understood. The regulation at this level would provide a sophisticated cellular mechanism to selectively impact PPP subcomplex functions, without causing deleterious effects via unselective modulation of PPP catalytic activities. Identification of such substrate recognition regulatory mechanism could also facilitate selective therapeutic PPP modulation approaches.

In this work we discovered structural mechanism inhibiting PP2A-B56 complex assembly and substrate recognition by a human oncoprotein CIP2A. We demonstrated that CIP2A can hijack B56α and PP2Ac from the PP2A-B56α trimer, function as a scaffold for the newly formed CIP2A-B56α-PP2Ac trimer, and subsequently mute substrate recognition by B56α (Fig. 6F). While the majority of the experiments were performed by using B56α protein, we validated the impact of N-terminal head mutant K21A also by using B56γ (Supplementary Fig. 5B, C). Together with the notion that the B56 family proteins are structurally very conserved on the regions implicated here in mediating CIP2A interaction, we postulate that the presented results can potentially be generalized over the B56 protein family. Based on the results, it can be envisioned how binding of CIP2A to the proximity of K227 on B56α would interfere with B56-PP2A-A interaction and thereby hijack B56 from PP2A-A (Fig. 5H, I). In this model, the local protein concentrations, and the active repulsion of PP2A-A from B56 upon CIP2A binding would be more critical than pure competition based on protein interaction affinities. Although the mechanism described here is clearly different from the mechanism previously described for TIPRL-mediated PP2A inhibition[56], they resemble each other regarding formation of an alternative trimer in which the inhibitor protein (CIP2A or TIPRL) substitutes for the canonical PP2A complex components. Whereas CIP2A replaces the A subunit from PP2A trimer, TIPRL, via several steps, hijacks the C-subunit to form a TIPRL-PP2Ac-α4 complex[56]. Together these studies indicate that hijacking of the PP2A complex components might be a commonly utilized mechanism for PP2A inhibition by endogenous inhibitor proteins.

Very importantly, we provide in cellulo validation for all major conclusions of the study. First, single point mutants of CIP2A head domain amino acids mediating B56α binding dramatically impacted CIP2A protein expression in cancer cells (Fig. 2). It is important to note that these results validate the relevance of the mutated N-terminal residues on the function of full-length CIP2A also containing the unstructured C-terminal tail. Second, AP-MS analysis of the B56α interactome confirmed that CIP2A over-expression expel A-subunit from B56α (Fig. 6B). Third, combination of two types of proteomics analysis with bioinformatics analysis confirmed that CIP2A inhibits B56α binding to LxxIxE motif substrates in cancer cells (Fig. 6 and Supplementary Data 4). Fourth, mutation of only one amino acid from the N-terminal head of CIP2A was sufficient to abrogate tumour growth of aggressive TNBC cell line (Fig. 3). Together, the discovered two-headed strategy by which CIP2A both disrupts functionality of the PP2A-B56α tumour suppressor complex, and simultaneously is itself protected from degradation, is likely to explain notable potency of CIP2A as a human oncoprotein. However, while PP2A-B56α obviously is an important target for CIP2A´s oncogenic activity, our data do not exclude other potential mechanisms by which CIP2A promotes oncogenesis.

The co-distribution of CIP2A cross-links with amino acids known to be critical for A subunit interaction with B56α provide mechanistic basis for competition between CIP2A and A subunit for B56α (Fig. 4A). We also provide substantial evidence to support the B56α muting

function for CIP2A. The cross-linking data, mutagenesis, and peptide competition experiments all support the conclusion that CIP2A N-terminal head binding to B56α sterically hinders LxxIxE motif substrate binding. However, unlike the bonafide B56α LxxIxE substrates, CIP2A binding to B56α is not dependent on the actual LxxIxE binding groove. Thereby CIP2A is not merely a pseudo-substrate for PP2A-B56α, but a steric inhibitor protein (Fig. 6F). Our results further implicate an important function for the LxxIxE groove region on stabilizing the PP2A trimer. Very interestingly, the recently identified CIP2A variant NOCIVA (NOvel CIP2A VAriant), that consists of the N-terminal CIP2A amino acids 1-545 fused to a unique C-terminal tail[57], contains all structural features identified here critical for CIP2A-B56α interaction. Therefore, it is likely that results of this study are relevant also for the understanding of the mechanism by which NOCIVA promotes myeloid cancer therapy resistance[57]. It is, however, evident that additional structural studies would be required to understand the molecular-level details of both CIP2A and NOCIVA interactions with PP2A-B56.

These results are likely to advance development of PP2A reactivating therapies. Our results reveal the CIP2A head domain as a critical structural region mediating PP2A-B56α inhibition, but also high CIP2A protein expression, as observed across human cancers[19,33]. Therefore, the results would pave the way for therapeutic strategies by targeting CIP2A. Based on near complete inhibition of tumorigenic properties of MDA-MB-231 cells by mutation of single amino acid K21 (Fig. 3), we propose that therapeutics preventing CIP2A head domain binding to B56α would constitute an efficient cancer therapy strategy by simultaneously preventing PP2A-B56α inhibition and by causing CIP2A protein destruction. Notably, the benefits of drug-induced target protein degradation are clear. Such an approach would not only inhibit CIP2A functions towards PP2A-B56α, but would remove any other CIP2A protein functions such as direct TopBP1 binding[24,26], and results in longer pharmacodynamic effects that are predicted to remain even after drug has been metabolized. Although the functional relevance was validated for TNBC cells here, therapeutic impact of inhibition of CIP2A protein expression has been validated by numerous studies across human cancer types[18,20,21,23,33,34]. CIP2A inhibition also sensitizes cancer cells to large number of cancer therapies[27,28,30].

In summary, this work describes a hijack and mute mechanism of multiprotein complex regulation that mediate inhibition of tumour suppressor PP2A-B56α. The results also provide clear cues for the development of novel targeted cancer therapeutics by revealing structural determinants behind oncogenicity of CIP2A. As CIP2A is also strongly implicated in Alzheimer´s disease via regulation of both Tau and APP[32,58], these results may have broad relevance to human diseases also beyond cancer.

## Methods

### Cloning

All CIP2A variants were generated by mutagenesis using polymerase chain reaction (PCR), with Phusion Green Hot Start High-Fidelity PCR Master Mix (F-5665, Thermo Scientific) and DpnI enzyme from Quick Change Lightning Site-Directed Mutagenesis Kit (210518-5, Agillent Technologies).

All plasmids for expression in mammalian cells were cloned in pcDNA3.1 vector using WT CIP2A(1-905) V5 His as a template. The following CIP2A variants were generated and using primers listed as Forward, F, and reverse, R and written in the 5'−3' direction: K8A (C TGC TTG GCG TCC TTG CTC CTG ACT GTC AG and CAA GGA CGC CAA GCA GGC AGT GGA GTC C), Q16E (GTC AGT GAA TAC AAA GCC GTG AAG TCA GAG and C TTT GTA TTC ACT GAC AGT CAG GAG CAA GG), K18A (CAG TAC GCG GCC GTG AAG TCA GAG GCG and CAC GGC CGC GTA CTG ACT GAC AGT CAG GAG), V20W (C AAA GCC TGG AAG TCA GAG GCG AAC GCC AC and C TGA CTT CCA GGC TTT GTA CTG ACT GAC AG), K21A (GCC GTG GCG TCA GAG GCG AAC GCC ACT C and CTC TGA CGC CAC GGC TTT GTA CTG ACT G), S22A (GTG AAG GCG

GAG GCG AAC GCC ACT CAG and CGC CTC CGC CTT CAC GGC TTT GTA CTG AC), E23K (G AAG TCA AAA GCG AAC GCC ACT CAG CTT TTG and GTT CGC TTT TGA CTT CAC GGC TTT GTA CTG), A24E (G TCA GAG GAG AAC GCC ACT CAG CTT TTG CGG and GGC GTT CTC CTC TGA CTT CAC GGC TTT GTA C), L33W (CGG CAC TGG GAG GTA ATT TCT GGA CAG AAA C and GA AAT TAC CTC CCA GTG CCG CAA AAG CTG AGT G), K40A (GGA CAG GCG CTC ACA CGA CTA TTT ACA TCA AAT C and G TGT GAG CGC CTG TCC AGA AAT TAC CTC CAA G), K647A (GAA ACA GCG GCT CTA GCC CTT GCA CAG G and C TAG AGC CGC TGT TTC CAA AAG ATC TTG TAG CC).

For expression in *Escherichia coli*, the following CIP2A variants were generated in pGEX4T1 vector: GST-CIP2A(1-330) A24E, GST-CIP2A(1-560) K8A and GST-CIP2A(1-560) K40A. B56α with R222E mutation was generated using the following primer pair GGA TTA GAA GCA TTC ATC AGA AAA CAA ATT AAC (F) and GAA TGC TTC TAA TCC AAG AAA TTT CCC ATA AAT TC (R), and using pGEX-B56α WT as a template. B56α with K181A/K217A8K227A mutations was generated using pEGFP-B56α WT as a template. The following primers to produce the B56α triple mutant were used: 5′- CCT GAT TTC CAG CCT AGC ATT GCA GCA CGA TAC ATT GAT CAG AAA−3′, 5′-TTT CTG ATC AAT GTA TCG TGC TGC AAT GCT AGG CTG GAA ATC AGG−3′, 5′-CTG TTC TGC ACC GAA TTT ATG GGG CAT TTC TTG GAT TAA GAG CAT TCA−3′, 5′-TGA ATG CTC TTA ATC CAA GAA ATG CCC CAT AAA TTC GGT GCA GAA CAG−3′, 5′-GGA AAT TTC TTG GAT TAA GAG CAT TCA TCA GAG CAC AAA TTA ACA ACA TTT TCC TCA G−3′, 5′-CTG AGG AAA ATG TTG TTA ATT TGT GCT CTG ATG AAT GCT CTT AAT CCA AGA AAT TTC C−3′.

B56α with K181A/K217A/K227A mutations was generated using pEGFP-B56α WT as a template. The following primers to produce the B56α triple mutant were used: 5′- CCT GAT TTC CAG CCT AGC ATT GCA GCA CGA TAC ATT GAT CAG AAA−3′, 5′-TTT CTG ATC AAT GTA TCG TGC TGC AAT GCT AGG CTG GAA ATC AGG−3′, 5′-CTG TTC TGC ACC GAA TTT ATG GGG CAT TTC TTG GAT TAA GAG CAT TCA−3′, 5′-TGA ATG CTC TTA ATC CAA GAA ATG CCC CAT AAA TTC GGT GCA GAA CAG−3′, 5′-GGA AAT TTC TTG GAT TAA GAG CAT TCA TCA GAG CAC AAA TTA ACA ACA TTT TCC TCA G−3′, 5′-CTG AGG AAA ATG TTG TTA ATT TGT GCT CTG ATG AAT GCT CTT AAT CCA AGA AAT TTC C −3′. B56α variant was generated using PCR based site-directed muta-genesis (Stratagene) with proofreading Pwo polymerase (Roche Applied Science) and complementary DNA oligonucleotide primers (Sigma Genosys) containing the desired point mutations. DH5α cells were routinely used for plasmid propagation. For DNA extraction and purification, the following commercial kits were used according to the manufacturer's instructions: NucleoSpin Plasmid (740588.50) and NucleoBond Xtra Maxi Plus EF (740426.50, both from Biotop).

All constructs were verified by sequencing (Finnish Microarray and Sequencing Centre, Turku Bioscience Centre, Finland, and FIMM Helsinki, Finland). B56α WT and B56α K181A/K217A/K227A mutants in pGEX4T2 vector were generated and sequenced by Genscript.

### Protein expression and purification for interaction assays

For expression in *E. coli*, all CIP2A variants were cloned in pGEX vector, which produces proteins as thrombin-cleavable amino-terminal GST-fusion proteins. BL21 cells were used for over-expression. The over-night bacterial culture was inoculated in LB media and incubated at 37 °C until $OD_{600}$ reached 0.6–0.9. Expression was induced with 0.2 mM isopropyl-β-D-1-thiogalactopyranoside (IPTG) for about 4 hh (H) at 23 °C. The bacterial pellets were collected by centrifugation at 6000 × *g* at 4 °C and stored at −20 °C until purification. Cells were lysed by sonication on ice, in a buffer consisting of 200 mM Tris pH 8, 500 mM NaCl, 2 mM dithiothreitol (DTT), 0.5% Tx-100, lysozyme (20 mg/150 mL) (Calbiochem 4403-1GM), and 1 × Pierce Protease Inhibitor Mini Tablets, EDTA-Free (Thermo Scientific A32955). Cleared lysate was incubated with glutathione agarose slurry (1:1 diluted with lysis buffer) at 4 °C (from 4 h to overnight), with gentle rotation

(Glutathione Sepharose 4B, 17-0756-01, GE Healthcare). Pelleted beads were washed extensively with washing buffer (same composition as lysis buffer, but without lysozyme), and then eluted using elution buffer: 100 mM Tris pH 8, 200 mM NaCl, 5 mM DTT, 0.1% Tx-100 and 20 mM Glutathione (L-Glutathione Reduced; Sigma-Aldrich G4251-5G). Samples were dialyzed using Snakeskin MWCO 10k (Thermo Scientific 88243), or Slide-A-Lyzer Dialysis Cassette MWCO 10 K (866383 Thermo Scientific), into a buffer containing 20 mM Tris pH 8, 150 mM NaCl, 2 mM DTT, 0.05% Tx-100 and 10% glycerol. If needed, the pooled fractions were concentrated using Amicon Ultra Centrifugal Filters (Merck Millipore), and concentration was determined by Coomassie staining (PageBlue Protein Staining Solution, Thermo Scientific 24630), using GST alone as internal standard.

B56α was also cloned in pGEX vector and expressed as above. Affinity purification using glutathione agarose was conducted over-night at 4 °C, with gentle agitation. Following extensive washes, GST tag was removed by overnight incubation at 4 °C with AcTEV protease (12575015, Invitrogen). TEV was inactivated with phenylmethane-sulfonyl fluoride (PMSF) at 1 mM final concentration, for 15 min at ambient temperature. R222E mutant of B56α was generated using WT B56α/pGEX as a template and purified under the same conditions used for WT protein.

The PP2A catalytic subunit PP2Ac was produced as a baculoviral protein as described in[41]. In brief, PR65 subunit was prepared from the pGEX vector containing TEV cleavage site. The protein was produced in *E. coli* BL21 strain induced by 0.2 mM IPTG when the O.D.600 was 0.6–0.8. The cells 30 were then shook at 23 °C and harvested after 3.5 h. The bacteria pellets were collected and lysed by sonication. The GST fusion protein was purified by Glutathione Sepharose 4B column. To produce the baculovirus for expression of the human catalytic Cα subunit, PP2Ac was cloned into the pFastBac HTb vector with an N-terminal HA-tag and a TEV cleavage site and was prepared from the Bac-to-Bac Baculovirus expression system (GibroBRL). Hi-5 cells in SF-900 II serum-free medium at a density of $2 \times 10^6$ cells/cm² were infected with fresh recombinant virus and incubated at 27 °C for 72 h. The cells were collected by centrifugation at 1000 × *g* for 20 min and pellets were washed with 50 mM Tris-HCl (pH 8.0), 100 mM NaCl, 5 mM β-mercaptoethanol with protease inhibitors (PMSF, leu-peptin, Ben-zamidine) and stored at −80 °C. Freeze/thaw and mild sonication was used to lysate cells, and cell debris was removed by centrifugation at 26000 × *g* for 1 h. The soluble fraction was filtered with 0.8 μm syringe filters and applied to the Ni-NTA affinity column pre-equilibrated with 50 mM Tris-HCl (pH 8.0), 100 mM NaCl, 3 mM β-mercaptoethanol. PP2Ac was eluted with elution buffer (50 mM Tris-HCl pH 8.0, 100 mM NaCl, 300 mM imidazole, 3 mM β-mercaptoethanol) and dialyzed overnight at 4 °C in 50 mM Tris-HCl pH 8.0, 100 mM 2 NaCl, 1 mM DTT. Dialyzed protein sample was incubated with TEV (1:20 w/w ratio) at room temperature overnight and re-applied to Ni-NTA to remove cleaved His-tag fragments. Flow-through fraction of second Ni-NTA was dialyzed again with 50 mM Tris-HCl pH 8.0, 25 mM NaCl, 1 mM DTT and applied into a Hi-trap HQ 5 ml column (Amersham Biotech) pre-equilibrated with 50 mM Tris-HCl pH 8.0, 1 mM DTT. Bound protein was eluted by one-step elution with 50 mM Tris-HCl pH 8.0, 500 mM NaCl, 1 mM DTT. The elution fraction was applied to a Superdex 200 size-exclusion column (Amersham Biotech) pre-equilibrated with 25 mM Tris-HCl pH 8.0, 50 mM NaCl, 1 mM DTT.

PP2A trimers and dimers were formed by incubating the A, B and C subunits in 1:1:1 stoichiometry (or 1:1 in case of AC dimer) for 1 h at room temperature (RT).

Plasmid encoding BubR1 fragment that interacts with B56α was pCOOL/ BubR1(647-720). BubR1 fragment's expression was induced with 0.1 mM IPTG for 4 H at 23 °C, and then overnight at 16 °C. For purification, the same protocol described for CIP2A was used.

The proteins used for XL-MS reactions were expressed and pur-ified following protocol outlined above, but the samples were finally

dialyzed into a buffer consisting of PBS pH 8, 2 mM DTT and 10% glycerol. For binding assays, samples were in dialysis buffer supplemented with 0.5% 3-[(3-cholamidopropyl)dimethylammonio]−1-propanesulfonate (CHAPS) (w/V) (BIMB1085 Apollo Scientific). Control samples included GST alone and samples without CHAPS.

## Chemical cross-linking (XL)

Free CIP2A and CIP2A-B56α complex were cross-linked (XL) with two different approaches[59,60]. All XL experiments were carried out at a final protein concentration of approximately 0.5 mg/mL and at scales of 50−100 µg total protein. For XL with disuccinimidyl suberate (DSS), samples were prepared in 1 × PBS buffer, pH 8.0, supplemented with 2 mM DTT and 10% glycerol. The CIP2A-B56α complex was assembled by mixing at a molar ratio of 1:2 (CIP2A:B56α) and incubation at RT for 60 min prior to cross-linking. DSS ($d_0/d_{12}$, Creative Molecules, 001S) was prepared as a 25 mM stock solution in anhydrous dimethyl formamide and added to the proteins to a final concentration of 100 or 500 µM. Samples were incubated for 30 min at 37 °C with mild shaking before quenching with ammonium bicarbonate to a final concentration of 50 mM. After further incubation for 30 min at 37 °C, samples were dried in a vacuum centrifuge. For cross-linking with pimelic dihydrazide (PDH $d_0/d_{10}$, Sigma-Aldrich, S364576 and 756903, respectively) and 4-(4,6-Dimethoxy-1,3,5-triazin-2-yl)−4-methylmorpholinium chloride (DMTMM, Sigma-Aldrich, 74104), the buffer was as above except the pH was adjusted to 7. PDH and DMTMM were added to final concentrations of 3.5 and 9.3 mg/mL, respectively, and the samples were incubated for 45 min at 37 °C with mild shaking. XL was stopped by passing the samples through Zeba gel filtration spin columns (ThermoFisher Scientific, 89882) and the filtrate was evaporated to dryness in a vacuum centrifuge.

## XL-MS sample processing for mass spectrometry (MS)

Dried, cross-linked samples were re-dissolved in 8 M urea to a protein concentration of 1 mg/mL. Tris(2-carboxyethyl)phosphine hydrochloride was added to 2.5 mM final concentration and samples were incubated for 30 min at 37 °C with mild shaking. Next, iodoacetamide was added to 5 mM and alkylation proceeded for 30 min at RT and protected from light. The urea concentration was reduced to approximately 5.5 M by diluting with 150 mM ammonium bicarbonate and endoproteinase Lys-C (Wako, 125-05061) was added at an enzyme-to-substrate ratio of 1:100. After incubation for 2.5 h at 37 °C, samples were further diluted to 1 M urea with 50 mM ammonium bicarbonate, trypsin (Promega, V5111) was added at an enzyme-to-substrate ratio of 1:50, followed by incubation at 37 °C overnight (typically 16 h). Digestion was stopped by acidification to 2% (V/V) formic acid and samples were purified by solid-phase extraction using Sep-Pak tC18 cartridges (Waters, WAT054960). Digested samples were fractionated by size-exclusion chromatography (SEC, GE Healthcare Superdex Peptide PC 3.2/30, 17-1458-01) on an Äkta micro FPLC system (GE). Three high-mass fractions were collected for MS analysis.

## Liquid chromatography-tandem mass spectrometry (LC-MS/MS) for XL-MS samples

For XL-MS, the experimental design was as follows: Three different samples (CIP2A only, CIP2A-B56a, CIP2A(1-330)-B56a), two different cross-linking chemistries (DSS, PDH + DMTMM), and two different DSS concentrations (100 µM, 500 µM). XL experiments were performed in single replicates for each scenario (8 different data sets = independent experiments), but fractionated into three fractions, and each fraction was analyzed by LC-MS in duplicate, resulting in considerable redundancy in the data (48 LC-MS runs in total).

SEC fractions were analyzed on an Easy nLC-1000 HPLC system coupled to an Orbitrap Elite mass spectrometer (both ThermoFisher Scientific). Peptides were separated by gradient elution on an Acclaim PepMap RSLC C18 column (150 mm × 75 µm, ThermoFisher Scientific,

164534) at a flow rate of 300 nL/min. The mobile phases were A = water/acetonitrile (ACN) /formic acid (98:2:0.15, V/V/V) and B = acetonitrile/water/formic acid (98:2:0.15, V/V/V), respectively, and the gradient was set from 9 to 35% B in 60 min. The Orbitrap Elite was operated in data-dependent acquisition mode with a precursor scan in the Orbitrap analyzer acquired at 120,000 resolution, followed by MS/MS scans of the ten most abundant precursors per cycle (charge state +3 or higher) in the linear ion trap at normal resolution. Fragmentation was induced by resonant excitation in the linear ion trap at 35% normalized collision energy. The isolation width was 2.0 m/z, and previously selected precursors were put on a dynamic exclusion list for 30 s.

## Data analysis for XL-MS

In a first step, a protein sequence database for the identification of XL peptides was generated by searching data from the analysis of the lowest mass SEC fractions for unmodified peptides using Mascot (version 2.5.1, MatrixScience). MS data in Thermo's proprietary raw format were first converted into the open mzXML format, the format used for xQuest (version 2.1.5)[60] searches described below. Conversion was performed using msconvert.exe, version 3.0.9393, part of the ProteoWizard toolbox, using the following options:−mzXML -−32−filter "peakPicking true 1−2". These settings correspond to 32-bit encoding and centroiding at the MS and MS/MS level. No further data processing was performed during conversion into mzXML format. mzXML files were further converted into mgf format using MzXML2Search, part of the Trans-Proteomic Pipeline (TPP v4.7 rev 0) using default parameters (MzXML2Search −mgf). The protein sequence database was UniProt/SwissProt 2015/09 without taxonomic restrictions. Mascot search parameters included: Enzyme, trypsin; maximum number of missed cleavages: 2; MS mass tolerance: 15 ppm; MS/MS tolerance: 0.6 Da; Instrument type: ESI-TRAP. Carbamidomethylation of cysteine was defined as a fixed modification, and oxidation of methionine was considered as a variable modification. Search results were filtered to 1% false discovery rate (FDR) according to Mascot's definition, and the top scoring contaminant proteins (Mascot score > 200) found in the samples were retrieved from UniProt and added to the sequences of the target proteins. For DSS cross-linking, the database contained 15 entries (two target proteins, 10 *E. coli* contaminants, two human keratins, bovine thrombin). For PDH/DMTMM XL, the database contained 23 entries (two target proteins, 14 *E. coli* contaminants, six human keratins and bovine thrombin).

To identify cross-linked peptide pairs, the mzXML files generated by msconvert were searched against these databases using xQuest (version 2.1.5)[60]. The following residues were specified as possible XL sites: For DSS, Lys; for PDH, Asp and Glu; for DMTMM, Asp with Lys or Glu with Lys. Specific search parameters were as follows: Enzyme, trypsin (= cleavage after K and R, but not before P); maximum number of missed cleavages (per peptide, excludes cross-linked K), 2; initial MS mass tolerance, 15 ppm; MS/MS tolerance, 0.2 Da for common ions and 0.3 Da for cross-linked ions; minimum peptide length, 4 amino acids. Carbamidomethylation of cysteine was defined as a fixed modification. The following post-search filters were applied: TIC ≥ 0.1; deltaS ≤ 0.9. The MS mass tolerance was further restricted to ± 5 ppm or less based on the actual mass error distribution at the time of data acquisition. MS/MS spectra of all remaining candidate identifications were manually evaluated and only identifications with at least four bond cleavages overall or three consecutive bond cleavages per peptide were retained. Finally, the following score thresholds were applied: For DSS, 16 for intra-protein cross-links and 17.5 for inter-protein cross-links; for PDH, 18 and 20, respectively; for DMTMM, 20 and 23, respectively. Precise error rate calculation from small databases is difficult, but we estimate that the FDR as determined by target/decoy searches is less than 5% at the non-redundant peptide pair level for each independent experiment. Cross-link identifications were visualized with xiNET[61]. The XL-MS data have been deposited to the ProteomeXchange Consortium

(http://proteomecentral.proteomexchange.org) via the PRIDE[62] partner repository with the dataset identifier PXD020636.

## In vitro protein interaction assays

For binding assays, all proteins, as well as the CIP2A peptide (aa.18-40), were used at 10 pmol. Samples were diluted in reaction buffer: 50 mM Tris HCl pH 7.5, 150 mM NaCl, 2 mM DTT, 0.2% Igepal, 10% glycerol, and incubated for 1 H at 37°. Reaction volume was 150 μL. Next, 5 μL input sample was withdrawn prior to adding 5 μL glutathione agarose (Glutathione Sepharose 4B,17-0756-01, GE Healthcare) (diluted 4 × in reaction buffer) to each sample. Precipitated complexes formed for 1 H at ambient temperature, by incubating samples with moderate rotation. The beads were washed by adding 250 μL of reaction buffer for total of four buffer exchanges and for 1 H at 4 °C, with moderate rotation. The bound complexes were eluted off the beads by adding 30 μL 2 × SDS-PAGE sample buffer, incubating for 10 min at 95 °C, then recovering the eluted proteins by centrifuging at $3000 \times g$ for 1 min. Eluted materials were resolved on 4–20% SDS-PAGE (Mini-Protean TGX Gels, Bio-Rad), transferred on PVDF membrane (Immobilon-P TRansfer Membrane IPVH0010 from Merck Millipore) and blotted as indicated. Images were quantified using Image J. For each sample, the signal from the V5 or B56α was first normalized against the eluted GST bait protein. Next, the value for CIP2A(1-560) was set as one and the values for the CIP2A mutants were adjusted accordingly. Data was plotted with GraphPad Prism6.1 showing mean + S.E.M. In the competition assay with B56α, GST-BubR1, CIP2A-(1-560)V5 His or CIP2A peptide (aa.18-40), B56α was first pre-incubated with CIP2A peptide or CIP2A protein for 30 min at 37 °C, then GST-BubR1 was added and reaction continued for another h at 37 °C. The subsequent steps were as described above.

## CIP2A peptide (aa18-40) synthesis

All chemicals were purchased from Merck/Sigma Aldrich. All MALDI spectra were recorded on a Bruker microflex. HPLC-MS analysis and HPLC purification were carried out on an Agilent Technologies 1260 Infinity I/II coupled to a 6120 Quadrupole LC/MS, as column for analytical runs was used a Macherey Nagel EC 250/4 Nucleodur 100-5 $C_{18}$ ec and for preparative runs Macherey Nagel VP 250/21 Nucleodur 100-5 $C_{18}$ ec. The solvents used for all analytical and preparative runs was a mixture of water containing 0.05% trifluoroacetic acid (TFA) and ACN containing 0.05% TFA. Peptide sequence: Biotin-K A V K S E A N A T Q L L R H L E V I S G Q K, Calculated mass: 2746.5 g/mol, Yield: 0.55%. The peptide was synthesized using solid-phase peptide synthesis on the automated peptide synthesizer MultiPep RSi from Intavis, using the fluorenylmethoxycarbonyl (Fmoc) protection strategy. Peptide elongation was carried out on a Lys pre-coupled 2-ClTrt resin, using HBTU/HOBt (2-(1H-Benzotriazol-1-yl)−1,1,3,3-tetramethyluronium-hexafluorophosphate/Hydroxybenzotriazole) double coupling (4 equiv of amino acid, 4 equiv of HOBt, 4 equiv of HBTU, 4 equiv of N-methylmorpholine (NMM) in dimethylformamide (DMF)) for 30 min, for the first 10 amino acids, then for 60 min for the remaining amino acids. The Fmoc deprotection was carried out using 20% piperidine in DMF. A capping step was performed after every coupling, using a 5% acetic anhydride ($Ac_2O$)/5% lutidine mixture in DMF. The cleavage cocktail contained 95% TFA, 2.5% triisopropylsilane (TIPS) and 2.5% water. The cleavage cocktail was applied for 4 h, followed by precipitation in cold ether ($Et_2O$). The peptides were analyzed by HPLC-MS and MALDI-MS and then purified by HPLC. The biotin coupling was carried out on the unpurified, resin bound peptide, right after the peptide synthesis. Biotin N-hydroxysuccimidine ester (17.1 mg, 50 μmol, 1 eq.) was dissolved in DMF (2 mL) and triethylamine (7 μL, 50 μmol, 1 eq.). The mixture was added to the resin-bound peptide and shook overnight at RT. The liquid was drained and the resin was washed twice with DMF and twice with dichloromethane (DCM). Peptide purity after HPLC was 87%.

## Size exclusion chromatography

Size exclusion chromatography was carried out using Superdex 5/150 column (GE Healthcare). The flow rate was 0.3 mL/min, and the column was operated at room temperature. The running buffer was 28 mM Tris pH 7.2, 150 mM NaCl, 0.05% NP-40, 1.25% glycerol, 2 mM DTT). All samples contained 50 pmol of each protein tested. The proteins were first let to form complexes by incubating them in the interaction buffer (50 mM Tris pH 7.5, 150 mM NaCl, 5% glycerol, 0.2% NP-40, 2 mM DTT) for 1 h at 37 °C. The total volume was 120 μl. The samples were centrifuged briefly at $11,000 \times g$ for 5 min before loading to the gel filtration column. In each run, 30 μl of the sample was injected to column. The total volume of the column is 3 mL.

## CIP2A co-immunoprecipitation assay

HEK-293-T cells were transiently co-transfected with 3 μg of V5-tagged CIP2A, or empty vector control, and 3 μg of GFP-tagged B56α variant, or empty vector control, using PEI transfection reagent according to a standard protocol. 1 million cells were plated on a 10 cm dish, one dish per experiment. 48 h post-transfection, the media was removed and the cells were rinsed with cold TBS, scraped on 1000 μL TBS, transferred to a clean Eppendorf tube and centrifuged for 20 s at $16,000 \times g$ at 4 °C. The cells were lysed in 200 μL NET buffer (50 mM Tris HCl pH 7.4, 150 mM NaCl, 15 mM EDTA and 1 % Nonidet P-40) supplemented with a protease and phosphatase inhibitor cocktail (Roche Applied Science), followed by a 15 min centrifugation at $13,000 \times g$.

For pull-down, the cell lysates were incubated with 20 μL GFP-trapping beads (ChromoTek) and 500 μL NENT100 buffer (20 mM Tris HCl pH 7.4, 1 mM EDTA, 0.1% Nonidet P-40, 25% glycerol, and 100 mM NaCl) containing 1 mg/mL bovine serum albumin (BSA), on a rotating wheel for 1 h at 4 °C. Then, the beads were washed four times with 500 μL NENT150 containing 150 mM NaCl.

Bound proteins were eluted by the addition of 2 × NuPage sample buffer (Invitrogen) and boiling. The eluted proteins were subsequently analyzed by SDS-PAGE on 4–12% (w/v) Bis-Tris gels (Invitrogen) and immunoblotting.

## PP2A activity assay

After GFP pull-down, beads were washed once more with 20 mM Tris HCl pH 7.4 and 1 mM DTT (Tris-DTT) and finally resuspended in 140 μL Tris-DTT solution. All assays were performed with 40 μL pulled GFP-tagged B56α diluted with an additional 40 μL Tris-DTT solution and 10 μL of 2 mM stock of K-R-pT-I-R-R phosphopeptide for 14–28 min at 30 °C.

The released free phosphate was determined by the addition of BIOMOL Green (catalog no. BML AK111-0250, Enzo). After 30 min of incubation at RT, absorbance at 620 nm was measured in a multichannel spectrophotometer. We subsequently obtained specific phosphatase activity by correcting the measured absorbance for the input of GFP-tagged B56α, as determined by immunoblotting with anti-GFP antibodies and signal quantification by ImageJ.

## Antibodies and Western blotting

The following antibodies were used: anti-V5 (monoclonal mouse (mM) Ab (E10/V4RR), Thermo Fisher Scientific, MA5-15253; 1:5000), anti-V5 (mM, Thermo Fischer Scientific, R960-25; 1:2000), anti-CIP2A (mM Ab(2G10-3B5), sc-80659, 1:1000), anti-GST (polyclonal rabbit (pR Ab), Thermo Fisher Scientific CAB4169; 1:10,000), anti-GST (pR Ab, Cell Signalling 2622 S; 1:10,000), anti-GST (mM Ab (B-14), sc-138; 1:10,000), anti-PR65 (pR Ab (H-300), sc-15355; 1:1000), anti-B56α (mM Ab (23), sc-136045; 1:500 and 1:5000), anti-β-Actin (mM Ab (C4), sc-47778; 1:5000 and 1:10,000), anti-B56γ (mM Ab (E6), 374380; 1:500), all from Santa Cruz Biotechnology), anti-c-Myc (mM Ab (9E10), sc-40, 1:1000), anti-PP2Ac (pR Ab, Cell Signalling 2038S; 1:1000 and 1:5000), anti-GAPDH (mM Ab (6C5), 5G4-6C5 Hytest, 1:10,000), anti-PR65 (clone C5.3D10, 1:1000) and anti-PP2Ac (clone F2.6A10, 1:500), both

generously supplied by Dr. S. Dilworth at Middlesex University, London, UK), anti-GFP (pRb Ab, 2555 S, Cell Signalling; 1:1000), anti-cleaved PARP (mM Ab(E51), ab32064 abcam; 1:1000), anti-Phospho-MEK1/2 (Ser217/221) (pR Ab, 9121S Cell Signallings; 1:1000), anti-phospho Vimentin (Ser39) (pR Ab, 13614 Cell Signalling; 1:1000), anti-HA tag (mRb Ab (C29F4), 3724 Cell Signalling; 1:200 and 1:1000).

Mini-PROTEAN TGX TM Precast Protein Gels 4–20% (BioRad) or SDS-PAGE on 4–12% (w/v) Bis-Tris gels (Invitrogen) were routinely used for SDS-PAGE electrophoresis. After electrophoresis, sample were transferred to PVDF membrane by wet electro-blotting (200 mA, 75 min) and probed for specific antibodies as indicated. The membranes were blocked in TBS supplemented with 3% BSA or 5% milk and 0.1% Tween20 for 1 h at RT and then incubated with the primary antibody overnight at 4 °C or 1–3 h at RT. After being washed in TBS with 0.1% Tween-20, the membranes were incubated at RT for 1 h with secondary antibodies. Secondary antibodies used were polyclonal goat anti-mouse immunoglobulin-HRP (P0447) and polyclonal swine anti-rabbit (P0399), both from Dako or polyclonal rabbit anti-mouse, P0260, Dako, and anti-rabbit IgG HRP-linked, 7074S, Cell Signalling: all used at 1:5000 dilution for 1 h at RT. For assays with B56γ and B56α(K181A/K217A/K227A), the following secondary antibodies (IRDye® 680RD Donkey anti-Mouse IgG Secondary Antibody, #926-68072 and IRDye® 800CW Goat anti-Rabbit IgG Secondary Antibody, #926-32211) were used at 1:10,000 dilution in TBS supplemented with 2% BSA and 0.2% Tween20 for 1 h at RT. To visualize antibody-antigen complexes, Pierce ECL Western Blotting Substrate (32106, Thermo Scientific) was used and incubated with the membranes for 1 min. The following films were used: Fuji Medical X-Ray Film (47410 19284, Fuji Film) and UltraCruz Autoradiography Film (sc-201697, Santa Cruz Biotechnology) and developed on an ImageQuant LAS 4000 system (GE Healthcare) with a WesternBright ECL detection kit (Advansta). Page Ruler Prestained Protein Ladder (26616 Thermo Scientific) was used as a protein size reference. All densitometric quantifications were done using ImageJ. Alternatively, Odyssey® CLx Imaging System was used and analyzed using ImageJ-win64 Fiji.

## Affinity purification coupled with mass spectrometry (AP-MS) analysis of B56α interactome

Murine NIH3T3 cells, stably expressing V5-tagged CIP2A WT or empty vector control, were transfected with GFP-B56α. Cells were plated on a 15 cm scale, with five dishes set for each condition, per experiment. Cells were transiently transfected with GFP-B56α-encoding plasmid using JetPrime, according to the manufacturer's instructions. Per 15 cm dish, 12 µg of DNA was used, and JetPrime was used at 2:1 ratio (24 µL per 15 cm dish). 24 h post-transfection, the media was removed and the cells were rinsed twice with cold PBS, scraped on 500 µL PBS, transferred to a clean Eppendorf tube and centrifuged for 20 s at 13,200 rpm at 4 °C. For AP-MS, the cell pellets were lysed in 200 µL NET buffer (50 mM Tris HCl pH 7.4, 150 mM NaCl, 15 mM EDTA, and 1% Nonidet P-40) supplemented with a protease and phosphatase inhibitor cocktail (Roche Applied Science), followed by a 15 min centrifugation at 13,000 × g.

For pull-down, the cell lysates were incubated with 60 µL GFP-trapping beads (ChromoTek) and 3 mL NENT100 buffer (20 mM Tris HCl pH 7.4, 1 mM EDTA, 0.1% Nonidet P-40, 25% glycerol, and 100 mM NaCl) containing 1 mg/mL bovine serum albumin (BSA) on a rotating wheel for 1 h at 4 °C. Then, the beads were washed twice with 500 µL NENT100 and twice with 500 µL NENT300 containing 300 mM NaCl. The B56α-GFP-trapped complexes were washed three times with 500 µL 50 mM Tris (pH 7.7) containing 50 mM NaCl, followed by three washes with 500 µL 200 mM ammonium bicarbonate (NH₄HCO₃). The traps were subsequently subjected to an overnight on-bead trypsin digestion at 37 °C in the presence of 200 mM NH₄HCO₃, 5% acetonitrile (ACN), 0.01% ProteaseMax (Promega) and 0.5 µg of trypsin (Pierce). The resulting peptides were desalted with C18 ZipTip pipette tips (Millipore) and subjected to high-resolution LC-MS/MS.

Six samples were analyzed of which three (biological) replicates originated from murine NIH3T3 cells stably expressing V5-tagged CIP2A WT and three (biological) replicates originated from corresponding empty vector controls, both transfected with GFP-B56α. No technical replicates were executed.

For UPLC separation, an Ultimate 3000 UPLC system (Thermo Fisher Scientific) equipped with an Acclaim PepMap100 pre-column (C18, particle size 3 µm, pore size 100 Å, diameter 75 µm, length 20 mm, Thermo Fisher Scientific) and a C18 PepMap analytical column (particle size 2 µm, pore size 100 Å, diameter 50 µm, length 150 mm, Thermo Fisher Scientific) using a 40 min linear gradient (300 nL/min) of 0–4% buffer B (80% ACN, 0.08% FA) for 3 min, 4–10% B for 12 min, 10–35% for 20 min, 35–65% for 5 min, 65–95% for 1 min, 95% for 10 min, 95–5% for 1 min, and 5% for 10 min, was used.

The Orbitrap Elite Hybrid Ion Trap-Orbitrap mass spectrometer (Thermo Fisher Scientific) was operated in positive ion mode (nanospray voltage 1.8 kV, source temperature 275 °C) in data-dependent acquisition mode with a survey MS scan at a resolution of 60,000 (FWHM at $m/z$ 200) for a mass range of $m/z$ 375–1500 for precursor ions, followed by MS/MS scans of the top 20 most intense peaks above a threshold count of 500 with charge +2 or higher. Low resolution CID MS/MS spectra were acquired in rapid CID scan mode using a normalized collision energy of 35 eV and an isolation window of $m/z$ 2.0. Dynamic exclusion was set to 10 s.

Raw MS data were converted into.mgf files and subjected to database searching using Proteome Discoverer software (version 1.4, Thermo Fisher Scientific). Peptides were identified by MASCOT (version 2.2.06, Matrix Science) using UniProt Mus musculus (91781 sequences; 28/09/20×18) and UniProt Homo sapiens (173330 sequences; 25/03/2020) as databases. The following MASCOT search parameters were used: Trypsin/P, oxidation (M) as variable modification, two missed cleavages, peptide tolerance 10 ppm for MS and 0.5 Da for MS/MS. Progenesis software (version 4.1.4832.42146, Nonlinear Dynamics) was used for relative quantification of peptides. Proteome Discoverer software (version 1.4, Thermo Fisher Scientific) was used for peptide annotation and peptide validation, the latter using the Percolator node (max delta Cn=0; max rank =1). Only peptides with a $q$ value <0.01 in any of the conditions (combined CTRL replicates versus combined V5-CIP2A replicates) were considered. Qlucore Omics Explorer (version 3.6.) was used to execute a two-sided t-test ($p < 0.05$) using log2 of protein abundances, normalized to the protein abundance of B56α in the CTRL_1 sample, as input.

## Cell culture and siRNA transfections

All cell lines 22RV1 (ATCC: CRL-2505), MDA-MB-231 (ATCC: CRM-HTB-26), HEK-293-T (ATCC: CRL-3216) and NIH-3T3 (ATCC: CRL-1658), were cultured in a humidified incubator maintained at 37 °C and 5% CO₂. 22RV1 cells were cultured in RPM1-1640 media (R5886-500ML, Sigma Life Science) supplemented with 10% (V/V) fetal bovine serum (FBS), 0.5% (V/V) penicillin/ streptomycin (10,000 U/10 mg per mL, Sigma) and 2 mM L-Glutamine (Biowest). MDA-MB-231 cells were maintained in DMEM (D6171-500 ML, Sigma Life Science) and supplemented as above. Cells were routinely passaged 2–3 times per week and regularly tested for mycoplasma; no contaminated cell lines were used. For siRNA transfections, Lipofectamine RNAiMAX (Thermo Fisher Scientific) was used following the manufacturer's instructions. $1 \times 10^5$ cells were seeded on a 6-well plate day before transfection to reach 60–70% confluency. Cells were then transfected with 10 pM siRNA and 7.5 µL RNAiMAX per well and assayed 48 h after transfection. Sequence of CIP2A siRNA was: 5′- CUG UGG UUG UGU UUG CAC U −3′[21].

## CRISPR/Cas9-mediated CIP2A mutagenesis

CIP2A_K21A mutagenesis were performed by CRISPR/CAS9 genome editing using the Alt-R CRISPR-CAS9 system (Integrated DNA Technologies, IDT) and Lonza 4D-Nucleofector X Unit (Lonza). crRNAs and

homology directed repair (HDR) template for CIP2A_K21A mutation were designed by the CRISPR Design tool on Benchling website (www.benchling.com) and the corresponding Alt-R CRISPR-Cas9 crRNAs (crRNA) were ordered from IDT. Cas9 RNP complexes were assembled immediately before nucleofection, by resuspending each RNA (crRNA and tracrRNA) or ssDNA oligonucleotides (HDR templates) in the appropriate volume of IDT Nuclease-Free Duplex Buffer. The final concentration of oligonucleotides should be 100 µM.

The crRNA was hybridized with Alt-R CRISPR-Cas9 tracrRNA (tracrRNA, IDT) by mixing 120 pmol of crRNA with 120 pmol of tracrRNA in 2.6 µL of CAS9 buffer (20 mM HEPES (pH 7.5), 150 mM KCl, 1 mM MgCl$_2$, 10% glycerol and 1 mM TCEP), incubating the mixture at 95 °C for 5 min and then letting the mixture cool to RT on benchtop (5–10 min). 100 pmol of Alt-R® S.p. Cas9-3NLS (IDT) in 3.36 µL of CAS9 buffer was slowly added to the crRNA:tracrRNA duplex and the subsequent solution was incubated for 20 min at RT to allow ribonucleoprotein complex (RNP) formation. The RNP complex and 1 µL of HDR template was then added to 20 µL of cell suspension containing 100,000 cells suspended in Nucleofector SE Cell line solution (Lonza, cat. V4XC-2032), mixed and 20 µL of the cell/RNP mix was pipetted into one well of a Nucleocuvette Strip (Lonza, cat. V4XC-2032). The reaction mixtures were nucleofected using MDA-MB-231 cell line specific programme in the 4D-Nucleofector, and finally transferred to 6-well plates. After 72 H, the nucleofected cells were single-cell cloned, and knock in (CIP2A_K21A) efficiency was analyzed from the single-cell clones by T7 endonuclease assay and Western blotting. The following are the crRNA and HDR template used to create MDA-MB-231 CIP2A_K21A, written in the 5′–3′ direction:

crRNA1: CUU UGU ACU GAC UGA CAG UCG UUU UAG AGC UAU GCU.

HDR Template 1: ATG GAC TCC ACT GCC TGC TTG AAG TCC TTG CTT CTG ACT GTC AGT CAG TAC AAA GCC GTG GCC TCA GAG GCG AAC GCC ACT CAG CTT TTG CGG CAC TTG GAG.

crRNA2: GUA CAA AGC CGU GAA GUC AGG UUU UAG AGC UAU GCU.

HDR Template 2: ATG GAC TCC ACT GCC TGC TTG AAG TCC TTG CTC CTG ACT GTC AGT CAG TAC AAA GCC GTG GCC AGT GAG GCG AAC GCC ACT CAG CTT TTG CGG CAC TTG GAG.

Genomic DNA was extracted from control and Cas9/sgRNA transfected MDA-MB-231 cells using the QuickExtract DNA Extraction solution (Lucigen) according to the manufacturer's protocol. Primers were designed to amplify a ~600 bp fragment surrounding the mutation site. CIP2A genomic primers: forward 5′-GGA AGC TTC TGA GAG CGA-3′ and reverse 5′-TGA CTT TCC GTC ACT GAG AAT A-3′CR The genomic loci of interest were amplified by PCR using 2× High-Fidelity PCR Master mix (NEB). To assess the gene editing efficiency, the T7 Endonuclease assay was used. Briefly, 200 ng of purified PCR product was diluted in T7EI reaction Buffer (IDT) and re-annealed using the following conditions: denaturation at 95 °C for 5 min, re-annealing by reducing the temperature to 85 °C at a rate of 2 °C per second, then from 85 to 25 °C at a rate of 0.1 °C per second, and a final hold at 4 °C. 2 µL of T7 Endonuclease I (T7EI) (IDT) enzyme was added to the annealed PCR products and incubated at 37 °C for 2 h. The T7EI digestion products were visualized by running on an Agilent Bioanalyzer DNA 1000 Chip (Agilent Technologies). Successful editing was determined by the presence of T7EI cleaved products in the Cas9/sgRNA transfected cells compared to control cells. Single cell clones were selected and sent for Sanger sequencing for further validation and analysis.

### Analysis of protein expression of CIP2A(1-905) WT V5 and its variants in mammalian cells

Cells were plated in 12 well-plate set-up and transfected using Lipofectamine 3000 or JetPrime transfecting reagents at 3:1 and 2:1 DNA: transfecting reagent ratios. The samples were collected 24 h post-transfection. 22RV1 cells were scraped on PBS, mixed with 2 × SDS-

PAGE sample buffer and incubated at 95 °C for 10 min. MDA-MB-231 cells were collected on lysis buffer consisting of 20 mM Tris pH 8, 150 mM NaCl, 2 mM DTT, 0.05 % Tx-100 and 1 × Pierce Protease Inhibitor Mini Tablets, EDTA-Free and lysed by sonication on ice for 2.5 min. Clarified lysates were loaded on 4–20% SDS-PAGE gels and analyzed by Western blotting using specific antibodies.

Images were quantified using Image J. For quantification, for each sample, the signal from the V5 antibody was first normalized against β-Actin or GAPDH. Next, the value for CIP2A(1-905) WT was set as one and the values for the CIP2A mutants were adjusted accordingly. Using GraphPad Prism (version 9.3.1), graphs were plotted and Two-sided t-test was calculated.

### Colony formation and anchorage independent growth assay

In a colony formation assay, 500 cells were seeded per well of a 6-well plate. After 10 days of growth, cells were fixed with cold methanol (100%), stained with 0.5% crystal violet for 15 min and washed with PBS to remove excess stain. The average colony area percentage was calculated using the 'ColonyArea' ImageJ plugin. In the anchorage-independent colony formation assay, 2000 cells were resuspended in growth medium containing 0.4% agarose (4% Agarose Gel, Thermo Fisher Scientific Gibco; top layer) and plated on a bottom layer containing growth medium and 0.8% agarose in a 12-well plate. After 14 days of growth, cells were stained with Nitro blue tetrazolium chloride and washed with PBS. Colonies were imaged using a Zeiss SteREO Lumar V12 stereomicroscope. Analysis was done using the ImageJ software. First, the background was subtracted using the rolling ball function with a radius of 50 µm, then auto-tresholding was applied to separate the colonies. Area percentage was calculated using the ImageJ built-in function 'Analyze Particles' with exclusion of particles smaller than 500 µm$^2$ that are not considered colonies.

### Phosphoproteome analysis of MDA-MB-231 CIP2A K21A clone

Phosphoproteomic analysis of MDA-MB231_control and MDA-MB231 CIP2A_K21A_Clone 1 was performed using biological triplicates for a total of six samples. The cells were lysed with 8 M urea (in 50 mM Tris-HCl, pH 8). Samples were reduced with 10 mM dithiothreitol (DTT, in 50 mM Tris-HCl, pH 8) at 37 °C for 60 min, and alkylated with 50 mM iodoacetamide (in 50 mM Tris-HCl, pH 8) at RT for 30 min in darkness. The reaction was quenched by addition of 45 mM DTT (in 50 mM Tris-HCl, pH 8). The urea concentration was diluted to under 1 M by addition of 50 mM Tris-HCl (pH 8). Proteins were digested using Sequencing Grade Modified Trypsin (Promega) in ratio 1:50 at 37 °C overnight. Digestion was quenched with addition of trifluoroacetic acid to pH 2 and desalted using micro column Sep-Pak tC18 100 mg 96-well plate (Waters) and dried in SpeedVac concentrator (Thermo Scientific). Peptide concentration was measured with Pierce BCA Protein Assay Kit (ThermoFisher Scientific) and 100 µg of peptides from each sample was labelled with TMTpro 16plex Label Reagent Set (ThermoFisher Scientific).

Samples were off-line fractionated with Agilent 1260 Infinity II series HPLC (Agilent Technologies) equipped with XBridge Peptide BEH C18 Column, 300 Å, 3.5 µm, 2.1 mm × 250 mm (Waters) at 200 µl/min flow. The gradient was scaled down from to match the lower flow rate and column diameter. 5% of fractions were pooled to 12 samples for total proteome analysis, and the remaining 95% to 6 samples for enrichment with High-Select™ Sachtopore-NP TiO2 beads (20um, 300 Å; ZirChrom) Phosphopeptide Enrichment Kit (ThermoFisher Scientific).

The LC-ESI-MS/MS analyses, for both total protein samples and phosphopeptide enriched samples, were performed on a nanoflow HPLC system (Easy-nLC1200, Thermo Fisher Scientific) coupled to the Orbitrap Fusion Lumos mass spectrometer (Thermo Fisher Scientific, Bremen, Germany) equipped with a nano-electrospray ionization source and FAIMS interface. Compensation voltages of −40 V, −60 V,

and −80 V were used. Peptides were first loaded on a trapping column and subsequently separated inline on a 15 cm C18 column (75 μm × 15 cm, ReproSil-Pur 3 μm 120 Å C18-AQ, Dr. Maisch HPLC GmbH, Ammerbuch-Entringen, Germany). The mobile phase consisted of water with 0.1% formic acid (solvent A) or acetonitrile/water (80:20 (v/v)) with 0.1% formic acid (solvent B). A 120 min 2-step gradient from 7 to 24% of eluent B in 62 min, to 36% of eluent B in 48 min, followed by a wash stage with 100% of eluent B was used to eluate peptides.

MS data was acquired automatically by using Thermo Xcalibur 4.4 software (Thermo Fisher Scientific). A Data dependent acquisition method (DDA) consisted of an Orbitrap MS survey scan of mass range 350–1750 $m/z$ followed by HCD fragmentation for the most intense peptide ions in a top speed mode with cycle time 1 sec for each compensation voltages with a resolution of 120k and a target value of 400k ions, and a maximal injection time of 50 ms, in profile mode. The ion charge stage of 2–6 were chosen for MS2 fragmentation. MS/MS spectra were acquired with a resolution of 50k, a target value of 50,000 ions, a maximal injection time of 86 ms, in centroid mode. Dynamic exclusion duration was 60 s. Protein identification and quantitation was performed by processing the raw data from all replicates using Protein Discoverer (PD) version 2.5 software (Thermo Scientific Inc. Germany) connected to an in-house server running the Mascot 2.7.0 software (Matrix Science). Data was searched against a SwissProt (version 2021_2) database using *Homo sapiens* taxonomy filter. The MS/MS spectra were deisotoped and deconvoluted by using the MS2 spectrum processor node in PD. A protein group list was generated from PD 2.5 software and the list was filtered to exclude contaminating proteins. Only proteins with $q$-value <0.01 (<1% FDR) as determined by percolator and detected in all replicates were used for further analysis.

The search criteria were as follows: trypsin as an enzyme; two missed cleavage sites allowed; carbamidomethylation of cysteine as a fixed modification; oxidation of methionine and phosphorylation of serine/threonine/ tyrosine and acetylation of the protein N terminus as variable modifications; peptide mass tolerance of 10 ppm; and MS/MS ion tolerance of 0.02 Da. For phosphorylation site localization, phospho-RS was enabled. Abundance values for peptides and proteins were calculated based on intensities of TMTpro reporter ions.

Statistical analysis were performed using two sample t-test and considered significant at $p < 0.05$. For phosphosite motif enrichment analysis, motifs were assigned using NetworKIN[63] with networkin_score cutoff 1.5. Enrichment was calculated between between sites that were differentially phosphorylated in the CIP2A K21A mutant and the rest of the identified phosphorylation sites using Fisher's exact test. P-values were adjusted using Benjamini-Hochberg method to account for multiple hypothesis testing. ptmRS score of at least 90 or greater was used to verify the modification site for motif enrichment analysis. ptmRS score was not considered necessary for reactome analysis.

### Orthotopic mammary fat pad xenograft
Female Athymic Nude mice (Hsd:Athymic Nude-*Foxn1^nu*, Envigo, Gannat, France), weighing between 19 and 24 g, were used at 6 weeks of age. The mice were housed in individually ventilated cages (IVC, Techniplast, Buguggiate, Italy; 3 to 4 mouse/ cage), under controlled conditions of light (12 hh light/ 12 hh dark), temperature (21 °C ± 3 °C) and humidity (55% ± 15%) in specific pathogen-free conditions at the Central Animal Laboratory, University of Turku (Turku, Finland). The mice were given irradiated soy protein-free diet (2920X, Envigo - Teklad Diets, Madison, WI, USA) and autoclaved tap water *ad libitum*. Mice were allocated according to body weights to three study groups MDA-MB-231 CIP2A_Control ($n = 7$), MDA-MB-231 CIP2A_K21A Clone1 ($n = 8$) and MDA-MB-231 CIP2A_K21A_Clone2 ($n = 8$), using published algorithm[64].

This study has been performed according to the guidelines following the EU legislation related to the use of animals for scientific purposes. National Animal Experiment Board of Finland authorized the animal studies with the license ESAVI/9241/2018 that were performed according to the instructions given by the Institutional Animal Care and Use Committees of the University of Turku.

The cells were suspended in the PBS (Dulbecco's Phosphate Buffered Saline, Biowest SAS, Nuaillé, France) described above at a density of $20 \times 10^6$ cells/mL. Thereafter, high protein concentration Matrigel™ (BD Biosciences, Bedford, MA, USA) was added (1:1) and a 150 μL aliquot of this suspension (1.5 million cells per mice) was inoculated orthotopicly (*o.t.*) under isoflurane (Isofluran, Baxter S.A., Lessines, Belgium) induced anesthesia to the left mammary fat pads of each mouse. In brief, a small incision was made midline to the nipple, and a tweezer was used to expose the mammary fat pad. A syringe with a 25 G needle was used to inoculate the cell suspension directly into the fat pad and the wound was closed with a suture. For pain relief, mice were injected subcutaneously (*s.c.*) with buprenorphine (0.1 mg/kg, Temgesic® 0.3 mg/mL, Reckitt Benckiser Healthcare, Hull, United Kingdom) and carpofen 5 mg/kg (Vet Rimadyl ® 50 mg/ mL, Pfizer SA, Louvain-La-Neuve, Belgium) before and after the operations.

Tumour growth was monitored three times a week by caliper measurement, and animals were weighed once a week. The volume of the tumours were calculated according to following formula: $W^2 \times L/2$ ($W$ = shorter diameter, $L$ = longer diameter of the tumour). Tumours were grown for 6 weeks or until tumours reached their maximum size (longer diameter reached 15 mm) and at the end of the study mice were sacrificed, and tumours were collected for further use.

All statistical analysis was performed two-sided. Student´s t-test was used for pairwise comparisons and Anova for xenograft data. Among the replicate values, the outliers were identified by Prism 9 (version 9.3.1) using 5% threshold with ROUT algorithm.

### Protein structure visualization
Protein structures figures were generated using Pymol or UCSF Chimera 1.11.2. PDB models used for figures are 5UFL[31], 6NTS[65], and 2IAE[41].

### Proximity ligation assay (PLA)
MDA-MB-231 cells were plated on coverslips in a 12-well plate format. Coverslips were pre-coated with poly-lysine (Sigma-Aldrich) and transfected using Jet prime (Polypus) at 2:1, according to the manufacturer's protocol. The following plasmid was used with the amount indicated: pCEP-4HA-B56α (1 μg)(generous gift from Dr. David Virshup). The assay was started 24 h after transfection. PLA kit from Sigma-Aldrich was used according to manufacturer's instructions. Primary antibodies were diluted in antibody diluent as follows: anti-HA (pR Ab (Y-11) sc-805)(1:200), anti-CIP2A ((2G10-3B5), sc-80659)(1:25) and incubated with the coverslips overnight at 4 °C. The slides were analyzed with laser scanning microscope Nikon Eclipse Ti2-E at 60 × magnification, and images were processed with Fiji-ImageJ. PLA was quantified using macro described in[61] and using the same parameters. Graphs were generated using GraphPad Prism, with Y axis representing the positive PLA signals per cell. Unpaired t-test with Welsh correction was used.

### Reporting summary
Further information on research design is available in the Nature Portfolio Reporting Summary linked to this article.

### Data availability
All mass spectrometry proteomics data have been deposited to the ProteomeXchange. Consortium via the PRIDE[62] partner repository and are available with following identifiers: Consortium via the PRIDE[66] partner repository and are available with following identifiers: PXD020636 (XL-MS proteomics data), PXD030297 (AP-MS proteomics data), PXD035179 (Phosphoproteomics data from K21A mutant cells). Published protein structures were obtained from the Protein Data

Bank (PDB) and are available under the accession codes 5UFL, 6NTS, and 2IAE. Source data are provided with this paper except for Fig. 4D–F and Supplementary Figs. 1B (partially), and 7 as original blots were lost due to inappropriate storage conditions. All other data are available from the authors on request. Source data are provided with this paper.

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

## Acknowledgements
Authors acknowledge the Turku Proteomics Facility, Cell Imaging and Cytometry, and Genome Editing Core at Turku Bioscience Centre, and Turku Centre for Disease Modelling at the University of Turku; all supported by Biocenter Finland. Taina Kalevo-Mattila is acknowledged for excellent technical support, as well as the entire Turku Bioscience personnel for excellent working environment. AL would like to thank Ruedi Aebersold (ETH Zurich) for access to infrastructure and instrumentation. The use of the facilities and expertize of the Protein Service core facility of the Tampere University and a member of Biocenter Finland, is gratefully acknowledged. The authors further want to thank Saverio Minucci for generously providing PLA reagents, Stephen Dilworth for PP2A antibodies, Wolfgang Peti for insightful structural biology advice, and Kari Kurppa for help with Crispr/Cas9 targeting. The work was supported by funding from Sigrid Juselius Foundation (J.W.), Academy of Finland (331237; J.W.)(331946; V.H.), Jane and Aatos Erkko Foundation (J.W.), Finnish Cancer Associations (J.W.), Finnish Cultural Foundation (K.P.), Albin Johanssons Foundation (K.P.), F.W.O. - Vlaanderen (Research Foundation – Flanders) (G0B0116N, G0B1719N, and 1S77521N) (V.J., J.D.O.), KU Leuven (C24/17/073) (V.J., R.D.), Belgian Foundation against Cancer (FA/2020/1330)(V.J.) and NIH NIGMS (R01GM144483)(RP).

## Author contributions
Study design: K.P., A.L., and J.W. Experimental work: K.P., A.L., N.G., J.D.O., R.D., A.A., R.H., N.H., O.K., J.O., Z.W., R.S., J.V., H.H., and J.M., and V.H. Supervision: M.K., W.X., V.J., J.W., and V.H. Reagents: M.K., J.N., and D.A. Data interpretation: K.P., N.G., J.D.O., R.D., R.H., J.O., R.V., O.K., R.P., W.X., V.J., J.N., A.L., and J.W. Manuscript writing: K.P., J.D.O., V.J., A.L., and J.W.

## Competing interests
J.W. is a consultant and scientific advisory member and K.P. a consultant for Anavo Therapeutics BV. Other authors declare no competing interests.
