## [Peer Review File · Nature Communications]

REVIEWER COMMENTS

Reviewer #1 (Remarks to the Author):

In this study by Pavic and colleagues, the authors use a combination of crosslinking mass spectrometry, phosphoproteomics, in vitro reconstitution, AP-MS and mutagenesis studies to shed additional light on how the known PP2A inhibitor, CIP2A, functions to abrogate PP2A-B56a signaling. Constitutive inhibition of PP2A is a known driver of many cancers, yet the molecular details for this remain elusive, which forms the basis for significance of the present work.

1. Experimental details for the phosphoproteomics experiments are not provided at all. In addition, all summary data from complete experiments should be provided as Supplementary Data tables in which peptides were identified by LC-MS/MS-based approaches (identification scores, etc.). Finally, although it appears that the XL-MS data have been uploaded to PRIDE, the phosphoproteomics data have not.
2. It is unclear why the authors did not attempt to express and crosslink additional fragments of CIP2A, including from the C-terminus (560 – 905). Does this region also participate in PP2A-B56a interactions?
3. Although the authors perform additional experiments to verify the extent of interactions between mutant forms of CIP2A and PP2A-B56a as a means to justify their use as selectively reactivating PP2A signaling, unfortunately these mutants exhibit significantly reduced expression. It has long been appreciated that reduced expression of CIP2A as a means to restore PP2A activity has deleterious effects on cancer cells. It is likely that a MDA-MB-231 cell clone expressing a stable shRNA that downregulates CIP2A to the same extent, without mutating it, would result in similar effects on tumor progression. Thus, the K21A mutant clone experiment is as easily explained by reduced expression of CIP2A as it is by specific and selective PP2A-B56a binding. The much more compelling experiment would be to leverage the double K647A/K21A mutants to truly confirm that the mechanism by which CIP2A K21A rescues PP2A function is through loss of B56a binding, vis-à-vis identical expression but differential PP2A-B56a binding.

Resolving these issues would greatly strengthen a revised manuscript.

Reviewer #2 (Remarks to the Author):

The authors for this manuscript, “Structural mechanism for inhibition of PP2A-B56alpha and oncogenicity by CIP2A”, mapped the potential interaction surfaces between CIP2A and B56 using xl-ms and then performed mutagenesis analysis to identify mutations that either enhance or abolish the

interactions with B56 and showed the correlation of B56 interactions and the cellular protein level of CIP2A. The authors then characterized a mutation, K21A, that destabilizes CIP2A to an intermediate level in the tumorigenesis of a TNBC model. The effects of this mutation on interactions with B56 could not be directly characterized because this mutant protein could not be produced like other mutants. Based on the xl-ms data, the authors reasoned that CIP2A not only interacts with B56 on the surface of the holoenzyme but also interacts with B56 surface at the interface to the scaffold subunit. Next, the authors showed that CIP2A interacts simultaneously with B56 and PP2A catalytic subunit, independent of the scaffold subunit, and showed preliminary data that CIP2A displaces the scaffold subunit from the holoenzyme and forms a complex with B56 and the catalytic subunit. The results are interesting and suitable for publication in Nature Communications if the conclusions could be supported by more robust data as outlined below:

1. Chemical crosslink could capture both specific interactions and nonspecific transit interactions. Separating these two types of crosslinks in xl-ms is crucial. The authors showed that the intermolecular crosslinks could be detected between residues in the N-terminal head domain of CIP2A less than 20 Å apart to B56 residues scattered over a broad surface area up to 100 Å apart. How could a surface area of less than 20 Å in dimension interact with a broad surface area many folds bigger? Strategies to separate specific crosslinks and nonspecific crosslinks are crucial.
2. The authors performed mutagenesis analysis of CIP2A crosslinked residues and identified mutations that either enhance or disrupt the interactions with B56, such as K8A and A24E. How about mutations to the residues in B56 that are crosslinked to K8 or A24 in CIP2A?
3. The fact that CIP2A bearing the K21A mutation could not be produced like other mutants suggests that this mutation might directly interfere with protein folding, rather than reducing the interactions between CIP2A and B56. The effects of this mutation on tumorigenesis thus could not be correlated to the disruption of CIP2A-B56 interactions. On the other hand, K8A has a strong impact on CIP2A-B56 interactions, which also correlates well with its effects on the cellular CIP2A level. So why wasn't this mutation chosen for tumorigenesis studies?
4. The authors made an interesting observation on the ability of CIP2A to displace the scaffold subunit for interaction with B56 and the PP2A catalytic subunit in a single gel filtration study. This preliminary observation could be readily supported by more robust data, such as time-dependent and CIP2A concentration-dependent displacement of the scaffold subunit and extracting of B56 and PP2A catalytic subunit from the holoenzyme.
5. PP2A holoenzymes have nanomolar inter-subunit binding affinities. What are the binding affinities of CIP2A to B56 alone, PP2A catalytic subunit alone, or both? It would be extremely helpful to measure these values by ITC. Could the measured binding affinities correlate to the ability of CIP2A to displace the scaffold subunit?
6. Do CIP2A mutations that disrupt or enhance B56 interactions also disrupt or enhance the interactions of CIP2A to B56 and PP2A catalytic subunit?

Reviewer #3 (Remarks to the Author):

In this manuscript, Pavic et al report a mechanism for how the tumour promoter CIP2A inhibits PP2A-B56. Based on crosslinking mass spectrometry and biochemical studies, the authors demonstrate that CIP2A uses its head domain to bind to the B56 and catalytic subunits. This binding occurs via a surface on B56 that would otherwise bind to the scaffold subunit in the holoenzyme complex. The authors provide convincing evidence that CIP2A is able to displace the scaffold subunit from the holoenzyme complex, and at the same time obscure the channel on B56 that would otherwise bind to LxxIxE motifs and target PP2A to substrates. Therefore, CIP2A disrupts the holoenzyme and then blocks substrate recruitment. The effect of this “hijack and mute” model is substantiated using mass spec analysis of B56 pulldowns from cells that do/don't express CIP2A, with less scaffold and LxxIxE containing proteins precipitated in the presence of CIP2A. Finally, since CIP2A is an oncoprotein, the relevance of these results for the oncogenic effects of CIP2A are evaluated in cell models and xenografts. Guided by the structural data, the authors use mutagenesis to identify residues in CIP2A that likely mediate the B56 interaction, showing that one of these mutations (K21A) reduces CIP2A protein expression, as predicted, and also abolishes its ability to induce anchorage independent growth, suppress apoptosis and drive tumour growth in xenografts. Therefore, mutations that likely disrupt B56 interaction are linked to the well-established tumourigenic effects of CIP2A.

This is an elegant study that presents an intriguing new model for how CIP2A works to inhibit PP2A-B56. In my opinion, the real strengths are the molecular characterisation of CIP2A/PP2A-B56 binding, and the demonstration that this displaces the scaffold subunit from the holoenzyme and sterically blocks the LxxIxE pocket. This information will be very important moving forward, because it will help to understand how PP2A-B56 is inhibited in cells, and it will help to characterise the tumourigenesis effects of CIP2A, which could potentially lead to novel strategies to reverse these effects. The cancer data is fully consistent with this model, but on the other hand, this is not altogether surprising since mutations that abolish CIP2A binding also reduce CIP2A protein expression. So one cannot, on the basis of these data alone, fully tie down whether it is PP2A binding that is critical for the observed effects of CIP2A in tumourigenesis. That is certainly the most likely explanation given all the evidence, but new experiments would be needed to definitively confirm this in cells. I have provided some ideas in relation to this below, but I do not see these as essential experiments because they could be challenging, and they may be better reserved for a follow-on study that is focussed on the tumorigenesis aspect. In my opinion, that would not detract from this study which is clearly focussed on the mechanisms of PP2A-B56 interaction and inhibition by CIP2A. This is very nicely characterised using a clear and elegant set of structural and biochemical analyses, which reveal a very interesting and novel mechanism.

Main questions, comments and suggestions:

- The manuscript focusses exclusively on CIP2A mutants that abolish binding to B56, but have the authors also attempted to find B56 mutants that prevent CIP2A binding? I appreciate this is not an easy task, but it may be a powerful way to determine if all of the CIP2A effects are mediated via B56 binding. Currently, this is not possible because B56-binding mutants in CIP2A also reduce CIP2A expression. However, I would predict that mutations of just one B56 isoform to become CIP2A-independent may preserve total CIP2A levels better, because CIP2A could still bind the other isoforms. This may then allow the functional B56 isoform to recover phenotypes in the presence of CIP2A protein. The ability to rescue individual B56 isoforms from CIP2A inhibition would also be very powerful by dissociating isoform-specific effects on tumourigenesis.

- An alternative or complementary strategy is to use mutants that rescue CIP2A levels without increasing B56 binding: For example, as predicted for K8A+K674 (S5B,C).

- I think it would be valuable to try an IP from cells with the CIP2A mutations shown in S5C. One would predict the K8A+A24E rescues B56 binding, but K8A+K647A does not. In my opinion, this would strengthen the current hypothesis, because otherwise the data that K8A directly abolishes B56 interaction is quite weak. Similarly, the K21A used in the tumourigenesis studies is only presumed to affect B56 binding based on a reduction in CIP2A levels - can this be tested in prostate cancer cells which still retain some K21A protein level? Or alternatively, can it be combined with A24E or K647A, as above, to improve expression and allow clearer assessment of effects on B56 binding.

- K8A only marginally decreased in vitro binding (2E) but has a dramatic effect on protein stability in cells (2B) – can this be explained? I was initially left thinking that K8A may affect protein solubility independently of B56 binding, but then the rescue of expression on A24E+K8A is very convincing (2I). I wonder if differences here could be explained by the fact that in vitro binding assays use only the N-terminal half of CIP2A and the cellular assays use the full length protein? For example, perhaps these residues control intramolecular interactions with C-terminal regions, that either inhibit (K8A) or allow (A24E) B56 binding? One would presume that B56 binding is regulate in some way, and if this is intramolecularly, then cancer causing mutations most probably relieve that inhibition to drive PP2A binding/inhibition. So, perhaps this is worth considering with respect to Q16E as well? In relation to this, have the authors tried to use either A24E or Q16E to purify full length CIP2A with B56? If they release autoinhibition then this may be needed to drive complex assembly

Other minor comment:

- I found the manuscript too focussed on B56 alpha throughout, and I began questioning whether CIP2A would even bind to other B56 isoforms. I presume it would, given previous data showing that it binds to

B56 gamma, so I don't think this necessarily needs new experiments to address, but at least some mention of possible redundancy or isoform-specific effects would be good.

- Regarding the mass spec analysis in Fig.6b. Are ZFGP59 and GNG12 in a complex with CIP2A and B56? They seem to be the only other two protein that are consistently increasing in B56 pulldown after CIP2A overexpression.

- Does the CIP2A dimer bind two B56 molecules or one? It would help to discuss this with respect to the cross-linking MS data and the size exclusion chromatography.

- I think it should be clarified earlier in the results that CIP2A(1-560) is the N-terminal half of the protein, and that this is the only fragment that is soluble in vitro. I had mistakenly thought it was the full length, until I got to line 196 when this was clarified.

- It would help to highlight the actual cross-linked residues in figure S3.

- The following statement should be modified to state that this is consistent with the notion that CIP2A inhibits recognition of LxxIxE motifs.

Line 474: "Notably, 63% of these proteins had a candidate LxxIxE motif (Fig. 6C and Table S2) confirming that CIP2A inhibits recognition of LxxIxE-motif targets by B56 α also in cellulose."

Adrian Saurin

Structural mechanism for inhibition of PP2A-B56 α and oncogenicity by CIP2A

Karolina Pavic et al.,

Response to reviewers:

We are extremely grateful for reviewer's enthusiasm over our data and their constructive suggestions for strengthening the manuscript. We truly apologize for a long delay in returning our revised manuscript due to severe problems related to logistics and the research material availability. Whereas the early phases of revision experiments were directly affected by Covid restrictions, in the latter part we encountered severe problems in deliveries of many critical research reagents. As an example, we lost three times a dry-ice package containing proteins etc. as the courier packages were stuck in different locations in Europe. Further, the PLA reagents did not arrive even after 5 months, but we obtained left-over reagents from two separate labs in Europe. Lastly, although personally very happy news, the pregnancy of the first author, Dr. Karolina Pavic, banned her from any laboratory work based on local regulations in Luxembourg, for the last 3 months until re-submission.

However, regardless of these significant challenges, we have been able to respond to all reviewer questions and provide significant new data clearly strengthening the main conclusions of the study. All our responses and new data has been explained in detail below. In total the paper now contains 20 new data panels based on the revision experiments. Collectively, the study provides a long-sought mechanistic explanation how one of the most prevalent human oncoproteins inhibits its tumor suppressor target PP2A. In addition to knowledge specifically relevant to understanding and future targeting of CIP2A, the data also reveal unprecedented mode of PP2A complex regulation. We sincerely hope that the new revised version of the manuscript can now be accepted for publication in *Nature Communications*.

REVIEWER COMMENTS

Reviewer #1 (Remarks to the Author):

In this study by Pavic and colleagues, the authors use a combination of crosslinking mass spectrometry, phosphoproteomics, in vitro reconstitution, AP-MS and mutagenesis studies to shed additional light on how the known PP2A inhibitor, CIP2A, functions to abrogate PP2A-B56a signaling. Constitutive inhibition of PP2A is a known driver of many cancers, yet the molecular details for this remain elusive, which forms the basis for significance of the present work.

1. Experimental details for the phosphoproteomics experiments are not provided at all. In addition, all summary data from complete experiments should be provided as Supplementary Data tables in which peptides were identified by LC-MS/MS-based approaches (identification scores, etc.). Finally, although it appears that the XL-MS data have been uploaded to PRIDE, the phosphoproteomics data have not.

Author response: We apologize for not providing this information with original submission. The experimental details are now explained in the materials and methods section and the requested MS/MS identification data is presented as **New Table S4**. The mass spectrometry proteomics data have been deposited to the ProteomeXchange Consortium via the PRIDE partner repository with the dataset identifier PXD035179.

2. It is unclear why the authors did not attempt to express and crosslink additional fragments of CIP2A, including from the C-terminus (560 – 905). Does this region also participate in PP2A-B56a interactions?

Author response: We totally agree with the suggestion that the C-terminal tail of CIP2A must have functional role, but unfortunately ours (and others) attempts to express and purify full length CIP2A(1-905), or CIP2A tail fragment (561-905) that would entail the C-terminus, have not been successful. This is probably due to the fact that CIP2A (561-905) C-terminal fragment is predicted to be largely unstructured. However, our current data shows that the full-length CIP2A (i.e. containing the C-terminal tail) that is mutated on critical N-terminal head domain residues, loses almost completely its capacity to bind B56, and has reduced stability in cancer cells (**New Fig. 3B,C (see below)** and Fig. 2). This clearly indicates that the identified N-terminal head domain functions are also relevant in the context of full-length CIP2A. This is now better emphasized in the text (ln. 204-207, 268-270, 659,662).

Figure 3: (B) Proximity ligation assay for interaction between HA-B56 α and endogenous CIP2A. MDA-MB-231-Control and MDA-MB-231_K21A mutant cells transfected with HA-B56 α construct were subjected to PLA with anti-HA and anti-CIP2A antibodies. Red dot indicates the association between HA-B56 α and endogenous CIP2A proteins. Shown is a representative image from N=3 PLA experiments. (C) Quantification of PLA shown in (B) was done using automated macro detecting PLA signals and described in³⁹. Unpaired t-test with Welch correction * $p < 0.05$.

3. Although the authors perform additional experiments to verify the extent of interactions between mutant forms of CIP2A and PP2A-B56a as a means to justify their use as selectively reactivating PP2A signaling, unfortunately these mutants exhibit significantly reduced expression. It has long been appreciated that reduced expression of CIP2A as a means to restore PP2A activity has deleterious effects on cancer cells. It is likely that a MDA-MB-231 cell clone expressing a stable shRNA that downregulates CIP2A to the same

extent, without mutating it, would result in similar effects on tumor progression. Thus, the K21A mutant clone experiment is as easily explained by reduced expression of CIP2A as it is by specific and selective PP2A-B56a binding. The much more compelling experiment would be to leverage the double K647A/K21A mutants to truly confirm that the mechanism by which CIP2A K21A rescues PP2A function is through loss of B56a binding, vis-à-vis identical expression but differential PP2A-B56a binding. Resolving these issues would greatly strengthen a revised manuscript.

Author response: *Directly following the reviewer's suggestion, we created two independent CIP2A shRNA MDA-MB-231 clones that have approximately similar level of CIP2A protein expression inhibition than was observed in K21A mutant MDA-MB-231 CRISPR clones (New Fig. S3E). We also titrated CIP2A siRNA to the levels that inhibited CIP2A expression only partially in MDA-MB231 cells (New Fig. S3H). Both types of CIP2A hypomorph cells were however yet fully capable in forming colonies in soft agar (New Fig. S3E-J). This demonstrates that the total loss of soft agar growth potential of K21A mutant clones (Fig. 3D,E) cannot be explained solely by inhibition of CIP2A protein expression in these clones, but involves also the changed function of the remaining K21A mutated protein (Fig. 3A, New Fig. 3B,C). In fact, both shRNA clones showed rather higher colony growth potential indicating that during their selection they have developed compensatory strategies for anchorage-independent growth whereas such phenomenon is not seen with similarly long-term selected K21A mutant cells. These results also imply that inhibition of anchorage-independent growth by siRNA approach requires almost total depletion of CIP2A protein as was evidenced in the previous study where such results were published by using MDA-MB-231 cells (Come et al, Clinical Cancer Research, 2009).*

To further substantiate the requested link between the role of K21 on B56 interaction, and on malignant cell growth, we performed proximity ligation analysis (PLA) of full-length CIP2A and B56 proteins from the WT and CRISPR/Cas9 K21A mutated MDA-MB-231 cells. The data shows that although there was approximately 50 % of K21A mutant protein expressed in these cells as compared to WT cells (New Fig. S6D), the K21A mutated full-length CIP2A protein was totally incapable of associating with B56a in cellulo (New Fig. 3B,C, see the data above).

We opted not to perform the suggested transient CIP2A mutant overexpression studies as there could have been marked differences on heterodimerization of the exogenous CIP2A with the endogenous CIP2A (that would have been impossible to control) and thereby the results from these experiments might have been very difficult to interpret. On the other hand, the new data described above do now provide compelling evidence linking the N-terminal head domain mediated CIP2A-B56 interaction to the oncogenic function of CIP2A.

Reviewer #2 (Remarks to the Author):

1. Chemical crosslink could capture both specific interactions and nonspecific transit interactions. Separating these two types of crosslinks in xl-ms is crucial. The authors showed that the intermolecular crosslinks could be detected between residues in the N-

terminal head domain of CIP2A less than 20 Å apart to B56 residues scattered over a broad surface area up to 100 Å apart. How could a surface area of less than 20 Å in dimension interact with a broad surface area many folds bigger? Strategies to separate specific crosslinks and nonspecific crosslinks are crucial.

Author response: Thank you for pointing out an important issue that was clearly not explained well enough in the manuscript. When interpreting the cross-link data, it is important to realize that in the list of the cross-linked peptides, it is not possible to distinguish which cross-link is coming from which of the two CIP2A dimer monomers. Thereby, it is theoretically possible that whereas head domain of one CIP2A monomer binds to LxxIxE groove region of B56, the head domain of the other monomer binds to more C-terminal region of B56. We had considered speculating about this type of “fork” assembly of CIP2A dimer-B56 interaction in the original version of the manuscript, but decided that it was too speculative suggestion as we do not have any hard data to support this idea.

We also want to emphasize that we used cross-linking reagent DSS at two different concentrations (100 and 500 μM), which both rendered very similar cross-linking profile, meaning that increasing concentration of the cross-linking reagent did not lead to cross-linking artefacts/ cross-links generated due to over-excessive reagent used. We also obtained similar cross-linking profile using different chemistries (DSS and DMTMM). All these data clearly increase our confidence that the main conclusions from the XL-MS experiments are biologically relevant and not artefactual. It is of note that we also observed differences in the cross-linking profile of CIP2A(1-560) alone versus in complex with B56. This further indicates high degree of flexibility of CIP2A dimer in solution, potentially creating transient interaction both with itself and with any of its binding partners, but that upon stable interaction with B56, the structure of CIP2A is significantly stabilized. Altogether, and regardless of certain caveats of the technology, this first ever structural data of CIP2A-B56 interaction provides significant information about structural properties and nature of the CIP2A dimer.

It is important to notice that the XL-MS experiments, including all proper controls listed above, are yet only a screening approach to get the best estimate what might be the most important regions involved in CIP2A-B56 interaction. On the other hand, experimental dissection of the relevance of each identified cross-link would be an enormous task that is clearly subject for future studies. Regarding the current data, we have very robustly validated the importance of N-terminal head domain in mediating the interaction which is further stabilized by the C-terminal region of CIP2A 1-560, as shown also by our new data (**New Fig. 4D, see data below**). Thereby, we totally agree with the reviewer that the amino acids addressed in the current study most probably do not represent exhaustive list of amino acids participating on CIP2A-B56 interaction, and that, as a screening experiment, the XL-MS data may contain some cross-links that are not functionally relevant, but strongly believe that, as such, our data provides already very significant advance on understanding CIP2A function as B56-PP2A inhibitor. These points are now discussed in the paper (e.g. In. 137-143, and 147-150).

Figure 4: (D) GST pull-down assay for PP2A trimer-GST-CIP2A(1-560) interaction. Equal molar amounts of PP2A proteins were used in all the samples. The amount of CIP2A(1-560) protein was titrated against PP2A as indicated. The proteins were incubated for 1h at 37 °C.

2. The authors performed mutagenesis analysis of CIP2A crosslinked residues and identified mutations that either enhance or disrupt the interactions with B56, such as K8A and A24E. How about mutations to the residues in B56 that are crosslinked to K8 or A24 in CIP2A?

Author response: We appreciate this relevant question. Indeed, as we identify several amino acids on B56 that have not been previously functionally implicated on B56 and PP2A complex function, but were now found as CIP2A cross-linked residues, this data will be very important future resource for interpreting B56 function.

Experimentally demonstrating which individual CIP2A crosslinked residues are most important for CIP2A-B56 interaction stability would be enormous task, as there were dozens of CIP2A crosslinked sites on B56, localized in two separate regions of B56. However, we addressed this important future research question at least partially by newly mutating the three cross-linked residues neighboring the LxxIxE substrate binding groove on B56, K181, K217 and K227. Not surprisingly, these mutations did not significantly impact direct interaction with recombinant CIP2A in a GST-pulldown assay (**New Fig. S8B**). This could be explained by the multiple other CIP2A-B56 cross-linked sites potentially stabilizing enough the interaction under these *in vitro* experimental conditions. However, this B56 triple mutant did significantly impact the stability of the PP2A trimer in cellulo. Specifically, in B56 pull-down experiments, the B56 mutations very potent inhibited both B56-A and B56-PP2Ac interaction (**New Fig. 5H and S8C, see data below**) and resulted in loss of catalytic PP2A activity (**New Fig. S8C**) from the pulldown samples. Thus, these results demonstrate that in addition to its critical role in substrate recognition, lysine residues bordering the LSPI groove on B56 play a role in PP2A trimer stability. This could be best explained by hydrogen bond between D177 of the PP2A-A subunit and K227 of the B56 subunit as illustrated in the **New Fig. 5I (see data below)**. Importantly, these results also provide further support to our novel model how CIP2A impacts PP2A-B56 holoenzyme activity. Based on the results it is easy to envision how binding of CIP2A to K227 would interfere with B56-PP2A-A interaction and thereby “hijack” B56 from PP2A-A as pictured in the model figure 6G.

Figure 5: (H) GFP-tagged B56α WT or indicated triple mutant were expressed in HEK293T cells. The amount of PP2A-A subunit bound to GFP-tagged B56α upon GFP trap pull-down was quantified by anti-PP2A-A immunoblotting. Shown are the ratios of the quantified anti-PP2A-A signal versus the quantified anti-GFP signal, relative to the B56α WT (set at 100 % in each experiment). A one-sample t-test (compared with 100 %) was used to assess statistical significance (*p < 0.05; **p < 0.01). **(I)** Triple B56α mutant (K181A/K217A/K227A) exhibits loss of K227 (B56)-D117(PP2A-A) salt bridge. Interaction is indicated with black dashed line. D177 of PP2A-A shown in yellow sticks corresponds to PP2A-A-B56α complex and shown in palecyan sticks corresponds to PP2A-B56γ complex. Structure was generated in Pymol.

3. The fact that CIP2A bearing the K21A mutation could not be produced like other mutants suggests that this mutation might directly interfere with protein folding, rather than reducing the interactions between CIP2A and B56.

Author response: We apologize for unclear description but K21A protein purification was not even attempted in the original manuscript version. In request to reviewer's comment we cloned the K21A mutant protein expression construct and purified the protein. The K21A recombinant protein could be expressed and purified well and thereby against previous predictions this mutation does not seem to interfere with protein folding (See the Coomassie gel picture below for review purposes).

The effects of this mutation on tumorigenesis thus could not be correlated to the disruption of CIP2A-B56 interactions.

Author response: By using the newly expressed mutant protein, we could verify that it interacts less efficiently with B56 than the corresponding CIP2A 1-560 WT protein (**New Fig. S5B,C**). As a B56 protein, we used for these assays B56 γ as thereby this experiment also addressed one of the reviewer questions related to relevance of the model across B56 family members.

To further substantiate the requested link between the role of K21 on B56 interaction and malignant cell growth, we performed proximity ligation analysis (PLA) of full-length CIP2A and B56 proteins from the WT and CRISPR/Cas9 K21A mutated MDA-MB-231 cells. The data shows that although there was 50 % of K21A mutant protein expressed in these cells as compared to WT cells (**New Fig. S6D**), the K21A mutated full-length CIP2A protein was totally incapable of associating with B56 in cellulo (**New Fig. 3B,C, see data below**).

Together, these new data provide compelling further evidence linking the N-terminal head domain mediated CIP2A-B56 interaction to oncogenic function of CIP2A. These data also directly correlate the impact of K21A mutation on the disruption of CIP2A-B56 interaction to its effect on malignant growth as requested by the reviewer.

Figure 3: (B) Proximity ligation assay for interaction between HA-B56 α and endogenous CIP2A. MDA-MB-231-Control and MDA-MB-231_K21A mutant cells transfected with HA-B56 α construct were subjected to PLA with anti-HA and anti-CIP2A antibodies. Red dot indicates the association between HA-B56 α and endogenous CIP2A proteins. Shown is a representative image from N=3 PLA experiments. (C) Quantification of PLA shown in (B) was done using automated macro detecting PLA signals and described in³⁹. Unpaired t-test with Welch correction * p < 0.05.

On the other hand, K8A has a strong impact on CIP2A-B56 interactions, which also correlates well with its effects on the cellular CIP2A level. So why wasn't this mutation chosen for tumorigenesis studies?

Author response: The K8A mutant was not chosen for CRISPR/Cas9 targeting, as it was expected to be deleterious based on our previous experience that it is impossible to create single cell clones of cells with total or near total inhibition of CIP2A expression.

4. The authors made an interesting observation on the ability of CIP2A to displace the scaffold subunit for interaction with B56 and the PP2A catalytic subunit in a single gel filtration study. This preliminary observation could be readily supported by more robust data, such as time-dependent and CIP2A concentration-dependent displacement of the scaffold subunit and extracting of B56 and PP2A catalytic subunit from the holoenzyme.

Author response: *As requested, we now present new evidence demonstrating CIP2A concentration-dependent effect of CIP2A on displacement of the scaffold subunit, and extraction of B56 and PP2A catalytic subunit from the holoenzyme (New Fig. 4D). These experiments were performed pre-Covid, but were not included in the original submission. Consistently with the other data, when recombinant CIP2A was incubated together with preassembled PP2A-B56 trimer, CIP2A could efficiently interact with both B56 and PP2A-C, but no interaction between CIP2A and PP2A-A was observed. As an additional important new information, we demonstrate higher capability of CIP2A 1-560 to extract B56 from the holoenzyme than CIP2A 1-330 (New Fig. 4D), which is consistent with role of C-terminal interaction region in stabilizing the CIP2A-B56 interaction (PMID: 28174209, and Fig. 1C,D).*

Figure 4: (D) GST pull-down assay for PP2A trimer-GST-CIP2A(1-560) interaction. Equal molar amounts of PP2A proteins were used in all the samples. The amount of CIP2A(1-560) protein was titrated against PP2A as indicated. The proteins were incubated for 1h at 37 °C.

5. PP2A holoenzymes have nanomolar inter-subunit binding affinities. What are the binding affinities of CIP2A to B56 alone, PP2A catalytic subunit alone, or both? It would be extremely helpful to measure these values by ITC. Could the measured binding affinities correlate to the ability of CIP2A to displace the scaffold subunit?

Author response: *The approximate affinities of CIP2A to B56 proteins have been published previously (PMID: 28174209). However, as these were MST measurements, they might not be directly comparable with for example ITC values reported for PP2A holoenzyme inter-subunit binding affinities. We aimed to address this reviewer's question experimentally, but*

these experiments were particularly affected by the logistic problems described above. As known very well by the field, ITC requires very high amounts of protein. On the other hand, production of recombinant PP2A-C is very challenging and therefore previously in all this type of experiments we had relied on Dr. Xu's laboratory in Shanghai. However, due to Covid no shipments from China to Finland was possible so that samples would arrive quickly enough to be still intact.

Importantly, the new data related obtained by the newly generated B56 mutants and explained in response to reviewer's question 2 above, do answer partially also this reviewer question. The finding that CIP2A covers the region on B56 that is critical for B56-PP2A-A interaction (around K227 in B56a), indicates that CIP2A binding to B56 actively "rejects" the PP2A-A subunit and thereby the displacement of the PP2A-A subunit might not be mediated simply by competition between the protein interaction affinities. Also, as the local concentrations of proteins (in this case CIP2A, B56 and PP2A-A) have a huge impact on protein interactions in cells, we find simple comparison of affinities in vitro unlikely to provide full mechanistic picture of how CIP2A would "hijack" B56 from PP2A-A. This model is now carefully discussed in ln. 678-683.

6. Do CIP2A mutations that disrupt or enhance B56 interactions also disrupt or enhance the interactions of CIP2A to B56 and PP2A catalytic subunit?

Author response: *For the same logistic reasons as above, these experiments could not be performed. Although we appreciate the interest to identify regions on CIP2A mediating the PP2A-C interaction, we do not consider that information essential for the main conclusions of this manuscript. The tools generated in this study will however be essential and freely available for the research community (plasmids will be submitted to Addgene after acceptance of the paper) to address this question in the future studies.*

Reviewer #3 (Remarks to the Author):

This is an elegant study that presents an intriguing new model for how CIP2A works to inhibit PP2A-B56. In my opinion, the real strengths are the molecular characterisation of CIP2A/PP2A-B56 binding, and the demonstration that this displaces the scaffold subunit from the holoenzyme and sterically blocks the LxxlxE pocket. This information will be very important moving forward, because it will help to understand how PP2A-B56 is inhibited in cells, and it will help to characterise the tumourigenesis effects of CIP2A, which could potentially lead to novel strategies to reverse these effects. The cancer data is fully consistent with this model, but on the other hand, this is not altogether surprising since mutations that abolish CIP2A binding also reduce CIP2A protein expression. So one cannot, on the basis of these data alone, fully tie down whether it is PP2A binding that is critical for the observed effects of CIP2A in tumourigenesis. That is certainly the most likely explanation given all the evidence, but new experiments would be needed to definitively confirm this in cells. I have provided some ideas in relation to this below, but I do not see these as essential experiments because they could be challenging, and they may be better reserved for a follow-on study that is focussed on the tumourigenesis aspect. In my opinion, that would not detract from this study which is clearly focussed on the mechanisms of PP2A-B56 interaction and inhibition by CIP2A. This is very nicely characterised using a clear and

elegant set of structural and biochemical analyses, which reveal a very interesting and novel mechanism.

Author response: We are extremely grateful for reviewer's enthusiasm over our data and his/hers constructive suggestions for strengthening the evidence for direct relationship between the structural and cancer data. We agree that future work will be needed to understand fully the relationship between direct CIP2A-B56 interaction and the observed growth and signaling effects, but we have already strengthened that evidence in the revised manuscript by following experiments:

1. We verify that recombinant K21A protein interacts less efficiently with B56 than the corresponding CIP2A 1-560 WT protein (**New Fig. S5B,C**). We used B56 γ as a B56 protein for these assays and thereby this experiment also addressed reviewer's question related to relevance of the data across B56 family members.
2. To further substantiate the requested link between the role of K21 on B56 interaction and malignant cell growth, we performed proximity ligation analysis (PLA) of full-length CIP2A and B56 proteins from the WT and CRISPR/Cas9 K21A mutated MDA-MB-231 cells. The data shows that although there was 50 % of K21A mutant protein expressed as compared to WT protein (**New Fig. S3D**), the remaining K21A mutated protein was totally incapable of associating with B56 in cellulo (**New Fig. 3B,C, see data below**).
3. We created two independent CIP2A shRNA MDA-MB-231 clones that have approximately similar level of CIP2A protein expression inhibition to what was observed in K21A mutant CRISPR clones (**New Fig. S3E**). We also titrated CIP2A siRNA to the levels that inhibited CIP2A expression also partially (**New Fig. S3H**). Both type of CIP2A hypomorph cells were yet fully capable in forming colonies in soft agar (**New Fig. S3E-J**) demonstrating that total loss of malignant growth potential of K21A mutant clones cannot be explained only by the approximately 50 % inhibition of CIP2A protein expression in these clones.

Figure 3: (B) Proximity ligation assay for interaction between HA-B56 α and endogenous CIP2A. MDA-MB-231-Control and MDA-MB-231_K21A mutant cells transfected with HA-B56 α construct were subjected to PLA with anti-HA and anti-CIP2A antibodies. Red dot indicates the association between HA-B56 α and endogenous CIP2A proteins. Shown is a representative image from N=3 PLA experiments. (C) Quantification of PLA shown in (B) was done using automated macro detecting PLA signals and described in³⁹. Unpaired t-test with Welch correction * $p < 0.05$.

Together these new data provide compelling further evidence linking the N-terminal head domain mediated CIP2A-B56 interaction to oncogenic function of CIP2A.

Main questions, comments and suggestions:

- The manuscript focusses exclusively on CIP2A mutants that abolish binding to B56, but have the authors also attempted to find B56 mutants that prevent CIP2A binding? I appreciate this is not an easy task, but it may be a powerful way to determine if all of the CIP2A effects are mediated via B56 binding. Currently, this is not possible because B56-binding mutants in CIP2A also reduce CIP2A expression. However, I would predict that mutations of just one B56 isoform to become CIP2A-independent may preserve total CIP2A levels better, because CIP2A could still bind the other isoforms. This may then allow the functional B56 isoform to recover phenotypes in the presence of CIP2A protein. The ability to rescue individual B56 isoforms from CIP2A inhibition would also be very powerful by dissociating isoform-specific effects on tumourigenesis.

Author response: *We refer to our answer above explaining how malignant growth effects of K21A mutant cannot be explained solely by inhibition of CIP2A protein expression.*

*We agree with the reviewer that as there were dozens of CIP2A cross-linked sites on B56, experimentally demonstrating which individual CIP2A cross-linked residues are most important for CIP2A-B56 interaction stability would be an enormous task. However, to address this important future research question, we focused on the three crosslinked residues neighboring the LxxLxE substrate binding groove on B56, K181, K217 and K227. These three residues were simultaneously mutated to alanines. Not surprisingly, these mutations did not significantly impact direct interaction with recombinant CIP2A (**New Fig. S8B**), which could be explained by the above mentioned multiple other CIP2A-B56 cross-linked sites potentially stabilizing enough the interaction under these in vitro experimental conditions. However, these B56 mutations had an instrumental role in stability of PP2A trimer in cellulo, demonstrated by very potent inhibition of both B56-A and B56-PP2A-C interaction, and loss of catalytic PP2A activity from the B56 mutant protein pulldown samples (**New Figs. 5H, (see data below) and S8C**). These results demonstrate that, in addition to its critical role in substrate recognition, the LxxLxE groove region on B56 has also an important role in PP2A trimer stability. This could be best explained by hydrogen bond between D177 of the PP2A-A subunit and K227 of the B56 subunit, as illustrated in the **New Fig. 5I (see data below)**. Importantly, these results also provide further support to our novel model how CIP2A impacts PP2A-B56 holoenzyme activity. Based on the results, it is easy to envision how binding of CIP2A to K227 would interfere with B56-PP2A-A interaction and thereby “hijack” B56 from PP2A-A as pictured in the model figure 6I.*

Figure 5: (H) GFP-tagged B56α WT or indicated triple mutant were expressed in HEK293T cells. The amount of PP2A-A subunit bound to GFP-tagged B56α upon GFP trap pull-down was quantified by anti-PP2A-A immunoblotting. Shown are the ratios of the quantified anti-PP2A-A signal versus the quantified anti-GFP signal, relative to the B56α WT (set at 100 % in each experiment). A one-sample t-test (compared with 100 %) was used to assess statistical significance (*p < 0.05; **p < 0.01). **(I)** Triple B56α mutant (K181A/K217A/K227A) exhibits loss of K227 (B56)-D117(PP2A-A) salt bridge. Interaction is indicated with black dashed line. D177 of PP2A-A shown in yellow sticks corresponds to PP2A-A-B56α complex and shown in palecyan sticks corresponds to PP2A-B56γ complex. Structure was generated in Pymol.

- An alternative or complementary strategy is to use mutants that rescue CIP2A levels without increasing B56 binding: For example, as predicted for K8A+K674 (S5B,C).
 - I think it would be valuable to try an IP from cells with the CIP2A mutations shown in S5C. One would predict the K8A+A24E rescues B56 binding, but K8A+K647A does not. In my opinion, this would strengthen the current hypothesis, because otherwise the data that K8A directly abolishes B56 interaction is quite weak. Similarly, the K21A used in the tumourigenesis studies is only presumed to affect B56 binding based on a reduction in CIP2A levels - can this be tested in prostate cancer cells which still retain some K21A protein level? Or alternatively, can it be combined with A24E or K647A, as above, to improve expression and allow clearer assessment of effects on B56 binding.

Author response: *Please see our response above related to link between N-terminal binding mutants and cancer phenotypes. Collectively, the data demonstrates that K21A mutant binds less to B56 both in vitro and in cellulo. Further, all new data strongly supports our main conclusions that inhibition of the direct interaction between the N-terminal head of CIP2A and B56 is responsible for the observed pronounced cancer effects, but that cancer effects cannot be explained only by reduced CIP2A protein stability. Additionally, although the suggested overexpression experiments could have been informative, we were concerned that there could have been marked differences on heterodimerization of the exogenous CIP2A with the endogenous CIP2A (that would have been impossible to control) and thereby the results from these experiments might have been very difficult to interpret.*

- K8A only marginally decreased in vitro binding (2E) but has a dramatic effect on protein

stability in cells (2B) – can this be explained? I was initially left thinking that K8A may affect protein solubility independently of B56 binding, but then the rescue of expression on A24E+K8A is very convincing (2I). I wonder if differences here could be explained by the fact that in vitro binding assays use only the N-terminal half of CIP2A and the cellular assays use the full-length protein? For example, perhaps these residues control intramolecular interactions with C-terminal regions, that either inhibit (K8A) or allow (A24E) B56 binding? One would presume that B56 binding is regulate in some way, and if this is intramolecularly, then cancer causing mutations most probably relieve that inhibition to drive PP2A binding/inhibition. So, perhaps this is worth considering with respect to Q16E as well? In relation to this, have the authors tried to use either A24E or Q16E to purify full length CIP2A with B56? If they release autoinhibition then this may be needed to drive complex assembly

Author response: *Currently we cannot explain the quantitative difference in the in vitro and in cellulo impact of K8A on CIP2A. Interestingly, the new data with K21A mutant shows similar pattern where the impact is fairly modest in vitro, whereas full length CIP2A mutant protein is strongly impacted in cellulo. As the effect with both mutants is consistent with known outcome of inhibition of CIP2A-B56 interaction (PMID: 28174209), we find this data directly supportive of our main conclusions. The only speculative explanation for the difference would be that while every mutant CIP2A molecule can be presumed to be directed (upon lower affinity and dissociation from B56) to degradation in cellulo resulting in protein exhaustion, which would not be seen in in vitro model.*

We totally agree with reviewer's speculation about potential impact of intramolecular interactions of CIP2A monomers which is partly supported by the high number of intramolecular cross-links in the XL-MS data. The reviewer's excellent suggestions how to potentially address these questions in the future are greatly appreciated.

Other minor comment:

- I found the manuscript too focussed on B56 alpha throughout, and I began questioning whether CIP2A would even bind to other B56 isoforms. I presume it would, given previous data showing that it binds to B56 gamma, so I don't think this necessarily needs new experiments to address, but at least some mention of possible redundancy or isoform-specific effects would be good.

Author response: *We intentionally focused the paper of B56a, as we believe that each PP2A B-subunit should be considered as their own entity, and because all experiments were performed by using B56a protein. However, we agree that as we previously have shown that CIP2A also binds to B56g, it was relevant to address this issue. To this end, we performed the GST pulldown experiment addressing the recombinant CIP2A K21A mutant by using B56g instead of B56a as the B56 family member (**New Fig. S5B,C**). Further, we structurally modelled the potential impact of B56a K227A mutation on PP2A-A interaction also by using the structure of B56g (**New Fig. 5I**). Consistently with conserved effects between B56a and B56g in other assays, also the structural analysis demonstrates that the proposed PP2A-A repulsion model would be conserved between B56a and B56g (**New Fig. 5I**). In response to*

these valid comments we omitted the alpha symbol from B56 in the title of the paper to generalize the results across B56 family and also bring this issue up in the discussion.

- Regarding the mass spec analysis in Fig.6b. Are ZFGP59 and GNG12 in a complex with CIP2A and B56? They seem to be the only other two protein that are consistently increasing in B56 pulldown after CIP2A overexpression.

Author response: *We do not have any explanation why interaction of these two proteins with B56 is increased upon CIP2A overexpression and neither of them have been seen previously as either B56 or CIP2A interaction partners based on BioGrid database.*

- Does the CIP2A dimer bind two B56 molecules or one? It would help to discuss this with respect to the cross-linking MS data and the size exclusion chromatography.

Author response: *Based on Coomassie stains, one CIP2A dimer appears to bind one B56 (Fig. 4B). When interpreting the cross-link data, it is important to realize that in the list of cross-linked peptides it is not possible to distinguish which cross-link is coming from which CIP2A monomer. Thereby it is theoretically possible that whereas head domain of one CIP2A monomer binds to LSPI groove region of B56, the head domain of the other monomer binds to more C-terminal region of B56. We had considered speculating about this type of “fork” assembly of CIP2A dimer-B56 interaction in the original version of the manuscript but decided that as we do not have any hard data to support this idea it was too speculative suggestion.*

- I think it should be clarified earlier in the results that CIP2A(1-560) is the N-terminal half of the protein, and that this is the only fragment that is soluble in vitro. I had mistakenly thought it was the full length, until I got to line 196 when this was clarified.

Author response: *Thank you for pointing out this issue, that has now been fixed (see ln. 126).*

- It would help to highlight the actual cross-linked residues in figure S3.

Author response: *Thank you for pointing out this issue, that has now been fixed.*

- The following statement should be modified to state that this is **consistent** with the notion that CIP2A inhibits recognition of LxxIxE motifs.

Line 474: “Notably, 63% of these proteins had a candidate LxxIxE motif (Fig. 6C and Table S2) confirming that CIP2A inhibits recognition of LxxIxE-motif targets by B56 α also in cellulose.”

Author response: *Thank you for pointing out this issue, that has now been fixed.*

REVIEWERS' COMMENTS

Reviewer #1 (Remarks to the Author):

This reviewer thanks the authors for their efforts to clarify their study in light of reviewer comments and concerns. In general, they have done a good job of doing so, while apparently dealing with ancillary challenges that are unfortunately all too common in science, especially during the COVID-19 pandemic.

On the other hand, I remain unconvinced that the tumorigenesis data they produced using the K21A mutant is due solely to disruption of CIP2A/PP2A-B56. While the authors have collected additional data to qualify the context of that experiment, they have chosen to not leverage the potentially conclusive double K21A/K647A mutant as suggested by me and Reviewer #3 to really nail down the contribution of PP2A binding from CIP2A protein depletion/stability. While the binding experiments are important, the nature of tumorigenesis in this model remains unclear. I recommend either performing this experiment and reporting on the data, or removing it entirely from the manuscript.

Otherwise, the study sheds important light on an elusive interaction in phosphatase biology that many labs have attempted to observe with little in the way of conclusive results.

Reviewer #2 (Remarks to the Author):

The authors made important efforts in addressing the reviewer's comments. The work represents an important advance in the PP2A field and is suitable for publication in Nature Communications.

Reviewer #3 (Remarks to the Author):

The authors have satisfactorily addressed all of my questions and I have no further issues to raise. I feel that the manuscript has been improved after review, especially with the inclusion of the new B56a mutant data that provides a potential mechanistic explanation for how CIP2A hijacks B56 from the scaffold subunit.

I congratulate all the authors on a very nice study.

Adrian Saurin

Authors' response to reviewer 1:

In his/her original comments the reviewer suggested several different approaches to address the question related to relationship between decreased CIP2A protein levels and total inhibition of soft agar growth in K21A mutant cells. After careful consideration of the suggested approaches, we decided to opt out mutant overexpression strategy as we considered that there is a high risk that the suggested experiments with the mutants would produce rather confusing than conclusive results. This was due to impossibility to control homo and heterodimerization of exogenous and endogenous CIP2A. Instead, we chose to use alternative approach suggested by the reviewer i.e. CIP2A shRNA, and further substituted the evidence by transient siRNA experiments. Therefore, in our opinion we fully followed one of the reviewer's alternative suggestions how to address this question.

The new data by the chosen approach very clearly demonstrated that the impact of K21A mutation on transformed cell growth cannot be solely explained by decrease in CIP2A protein expression. Furthermore, the new PLA data clearly showed that the CIP2A-B56 interaction is inhibited in K21A mutant cells. Thus, we considered that we successfully responded to all of his/hers concerns.

To directly response to reviewer's comments, we already removed from the original revised version of the manuscript any claims that the growth effects seen with the K21A mutation would be solely due to inhibition of B56 interaction. We however have now even more clearly stated these caveats of the current data and toned down the related conclusions to reflect these limitations (ln.285-286).

" This demonstrates that the total loss of malignant growth potential of K21A mutant Crispr clones (Fig. 3F,G) cannot be explained only by partial inhibition of CIP2A protein expression in these clones, but it also involves both expression regulation and loss of CIP2A-B56 interaction"